# CodeT: Code Generation with Generated Tests

**Bei Chen**[*]**, Fengji Zhang**[*]**, Anh Nguyen**[*]**, Daoguang Zan, Zeqi Lin,**
**Jian-Guang Lou, Weizhu Chen**
Microsoft Corporation
{beichen, v-fengjzhang, anhnguyen, v-dazan,
zeqi.lin, jlou, wzchen}@microsoft.com

## Abstract

The task of generating code solutions for a given programming problem can benefit from the use of pre-trained language models such as Codex, which can produce multiple diverse samples. However, a major challenge for this task is to select the most appropriate solution from the multiple samples generated by the pre-trained language models. A natural way to evaluate the quality and correctness of a code solution is to run it against a set of test cases, but the manual creation of such test cases is often costly and time-consuming. In this paper, we propose a novel method, CodeT, that leverages the same pre-trained language models to automatically generate test cases for the code samples, thus reducing the human effort and increasing the coverage of the test scenarios. CodeT then executes the code samples using the generated test cases and performs a dual execution agreement, which considers both the consistency of the outputs against the generated test cases and the agreement of the outputs with other code samples. We conduct comprehensive experiments on four benchmarks, HumanEval, MBPP, APPS, and CodeContests, using five different pre-trained language models with varying sizes and capabilities. Our results show that CodeT can significantly improve the performance of code solution selection over previous methods, achieving remarkable and consistent gains across different models and benchmarks. For instance, CodeT improves the pass@1 metric on HumanEval to 65.8%, which represents an absolute improvement of 18.8% over the code-davinci-002 model, and an absolute improvement of more than 20% over the previous state-of-the-art results.

## 1 Introduction

Despite the remarkable progress in pre-training techniques for code generation, selecting a single correct solution from multiple candidates generated by large language models remains a hard problem. For instance, Codex (Chen et al., 2021), a state-of-the-art pre-trained language model for code generation, can achieve a pass@100 (pass if one or more among 100 generated solutions for a given problem can pass the corresponding test cases) of 77.4%, but a pass@1 (correct rate of a single solution) of only 33.5% on the HumanEval benchmark (Chen et al., 2021)[1]. This huge gap limits the practical usefulness of code generation models and motivates us to explore how to pick the correct or best solution from multiple candidates.

A straightforward way to verify the correctness of a solution is to execute it and check if it passes all corresponding test cases. This execution-guided approach has been widely adopted in various code-related tasks, such as code generation (Chen et al., 2021; Li et al., 2022b; Shi et al., 2022), code translation (Roziere et al., 2021), and program synthesis (Chen et al., 2018; Ellis et al., 2019). However, this approach relies heavily on the quality and quantity of test cases, which are often costly and time-consuming to create and maintain. Moreover, in real-world applications like Copilot[2], a code generation tool that assists developers in writing code, it is unrealistic to expect users to provide test cases for every problem they want to solve. Therefore, we propose to automatically generate test cases for arbitrary programming problems and use them to quickly verify any solution.

---

[*]The first three authors contributed equally.

[1]We report the results on the HumanEval benchmark with the Codex model code-cushman-001. More results with different models and benchmarks can be found in Section 4.1 and 4.2

[2]https://github.com/features/copilot

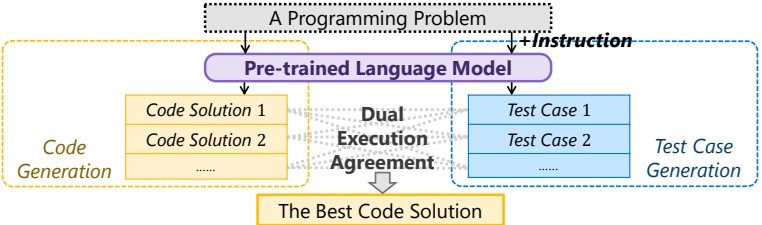

Figure 1: The illustration of CODET. Both the code solutions and the test cases are generated by the pre-trained language model. The best code solution is then selected by a dual execution agreement.

In this paper, we propose CODET: CODE generation with generated Test-driven dual execution agreement, as illustrated in Figure 1. First, we leverage the same pre-trained language model that generates code solutions, such as Codex, to generate a large number of test cases for each programming problem by providing an elaborate instruction as prompt. Next, we use a dual execution agreement approach inspired by the classical RANSAC algorithm (Fischler & Bolles, 1981). We execute each generated code solution on each generated test case, and iteratively find multiple groups of code solution and test case pairs. Each group, or consensus set, has solutions that pass the same test cases, indicating that they have the same functionality, even if they are different in implementation. We expect that a solution that passes more test cases is more correct, and that a solution that has more similar solutions, i.e., solutions in the same consensus set, is more consistent with the problem specification. So, we rank each consensus set by both the number of test cases and solutions in it, and choose the best solution from the highest-ranked consensus set.

Our method is simple and efficient, as it does not require any labelled data or additional rankers, but it achieves surprisingly exceptional performance. We evaluate our method on five different pre-trained language models for code generation: three OpenAI Codex models (Chen et al., 2021), INCODER (Fried et al., 2022), and CODEGEN (Nijkamp et al., 2022), as well as four established benchmarks for code generation: HumanEval (Chen et al., 2021), MBPP (Austin et al., 2021), APPS (Hendrycks et al., 2021), and CodeContests (Li et al., 2022b). The experimental results show that our method can effectively select the correct solution from multiple candidates, improving the pass@1 score significantly on all benchmarks in the zero-shot setting. For instance, CODET achieves improvements using code-davinci-002: HumanEval ($47.0\% \rightarrow 65.8\%$), MBPP ($58.1\% \rightarrow 67.7\%$), APPS INTRODUCTORY ($27.2\% \rightarrow 34.6\%$), and CodeContests ($0.7\% \rightarrow 2.1\%$). Moreover, when we combine code-davinci-002, the most powerful pre-trained model, and CODET, we outperform previous state-of-the-art methods by a large margin, e.g., HumanEval: $42.7\%$ (Inala et al., 2022) $\rightarrow 65.8\%$. We also conduct a thorough analysis to provide more insights. Our work is publicly available at `https://github.com/microsoft/CodeT`.

## 2 METHODOLOGY

The task of code generation is to solve a programming problem: generate *code solution* $x$ based on *context* $c$. As shown in Figure 2, context $c$ contains natural language problem description in the form of code comment, and a code snippet that includes statements such as imports and the function header. A code solution is a code snippet that solves the programming problem described in the context. Generally, we sample a set of code solutions, denoted as $\mathbf{X} = \{x_1, x_2, \cdots, x_N\}$, based on the context $c$ using a pre-trained language model $\mathcal{M}$, which can be formulated as $\mathbf{X} = \mathcal{M}(c)$. Our goal is to select the best code solution $\hat{x}$ from the set of generated code solutions $\mathbf{X}$, where $\hat{x}$ is the most likely solution to correctly solve the given programming problem. To this end, we propose CODET in the hope of unleashing the inherent power of the pre-trained language model $\mathcal{M}$. Specifically, we use $\mathcal{M}$ to generate test cases for the programming problem (Section 2.1), and then select the best code solution $\hat{x}$ based on a dual execution agreement (Section 2.2).

### 2.1 TEST CASE GENERATION

Besides generating code solutions, we also need to generate test cases to evaluate the correctness of the code solutions. A test case is a pair of input and expected output for the function defined in the

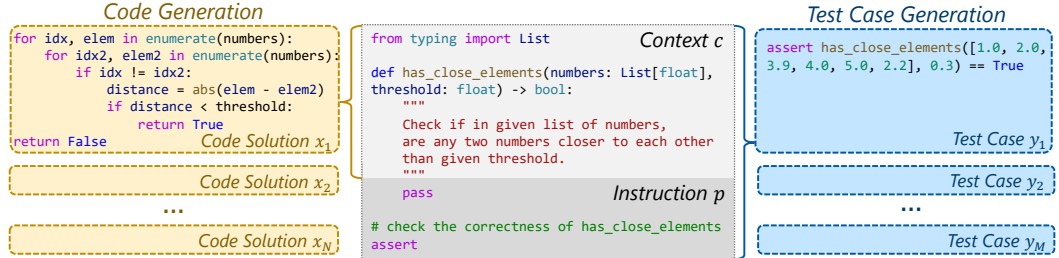

Figure 2: Code generation and test case generation: an example from the HumanEval benchmark. Example input-output cases are removed from the context.

context. For example, in Figure 2, a test case for the programming problem of checking whether there exist close elements in a list less than a threshold. To generate test cases, we use the same pre-trained language model $\mathcal{M}$ that we use for generating code solutions, but we add an *instruction* $p$ to the context $c$ as a prompt to indicate that we want test cases instead of code solutions. As shown in Figure 2, the instruction $p$ consists of three parts: (1) a "pass" statement as a placeholder of the function body, which signals that we do not need to generate code for the function, (2) a comment "check the correctness of [entry point]" to clarify the intention of generating test cases, where "[entry point]" is the name of the function, and (3) an "assert" statement to start the test case generation, which specifies the format of the test cases as input-output pairs.

We then feed the concatenated context and instruction, $\text{concat}(c, p)$, to the language model $\mathcal{M}$, and sample a set of test cases, denoted as $\mathbf{Y} = \{y_1, y_2, \cdots, y_M\}$, from the model output. The process of test case generation can be formulated as $\mathbf{Y} = \mathcal{M}(\text{concat}(c, p))$. The language model will try to complete the instruction by generating plausible input-output pairs for the function. Note that we remove all example input-output cases from the context $c$ before generating code solutions and test cases, to avoid exposing real test cases to the language model.

## 2.2 DUAL EXECUTION AGREEMENT

In this subsection, we explain how we select the best code solution $\hat{x}$ from the set of generated code solutions $\mathbf{X} = \{x_1, x_2, \cdots, x_N\}$, using the set of generated test cases $\mathbf{Y} = \{y_1, y_2, \cdots, y_M\}$ as a criterion. We can execute a code solution $x$ on a test case $y$, which means running the function defined by $x$ on the input part of $y$ and comparing the output with the output part of $y$. If the code solution $x$ can be executed without errors and the output matches the expected output, then we say the code solution $x$ can pass the test case $y$. Furthermore, we say there is a functionality agreement between two code solutions $x_i$ and $x_j$ if they can pass the same set of test cases in $\mathbf{Y}$. Our approach is based on the following assumptions: (1) the code solutions and the test cases are independently and randomly sampled from the pre-trained language model $\mathcal{M}$ given a certain programming problem, and (2) incorrect code solutions are often diverse, and the probability of having a functionality agreement between two incorrect code solutions by chance is very low. These assumptions are similar to those of the classical RANSAC algorithm (Fischler & Bolles, 1981), which is a robust method for finding consensus among noisy data. Inspired by RANSAC, we propose our approach CODET to perform dual execution agreement, which is an iterative approach as follows:

- We randomly select a pair $(x, y)$ from the set of all possible pairs $\mathcal{D} = \{(x, y)|x \in \mathbf{X}, y \in \mathbf{Y}\}$. We then try to execute the code solution $x$ on the test case $y$. If $x$ can pass $y$, then we say that the pair $(x, y)$ is a *hypothetical inlier*, because it hypothetically describes the correct functionality for the programming problem. Otherwise, we say that $(x, y)$ is an *outlier*, because it fails to describe the correct functionality. Figure 3 shows a simple example of the programming problem "return the square of a number". $(x_1, y_1)$ and $(x_3, y_2)$ are two of the hypothetical inliers, while $(x_1, y_4)$ and $(x_3, y_1)$ are two of the outliers.

- If $(x, y)$ is a hypothetical inlier, we collect all other pairs from $\mathcal{D}$ that agree with this hypothetical inlier, forming a set $\mathcal{S}$ called *consensus set*. To find the pairs that agree with $(x, y)$, we first find all test cases that $x$ can pass, denoted as $\mathcal{S}_y$. Then, we find all code solutions that can pass exactly the same test cases as $x$, denoted as $\mathcal{S}_x$. Finally, the con-

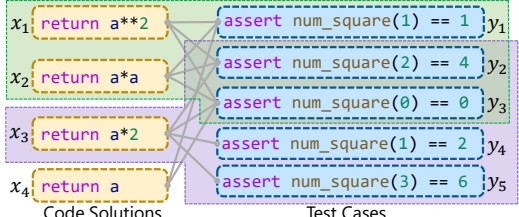

Figure 3: A simple example of the programming problem "return the square of a number". The gray line between $x$ and $y$ indicates that $x$ can pass $y$, i.e., $(x, y)$ is a hypothetical inlier. The green or purple box indicates a consensus set.

Table 1: Statistics of benchmarks: the total number of problems in the benchmark (*Problems*), the average number of ground-truth test cases per problem (*GT Tests*), and the number of sampling code solutions for each problem ($n$).

| | Benchmark | Problems | GT Tests | $n$ |
|---|---|---|---|---|
| | HumanEval | 164 | 7.77 | 100 |
| | MBPP | 427 | 3.1 | 100 |
| APPS | INTRODUCTORY INTERVIEW COMPETITION | 1,000 3,000 1,000 | 20.99 | 50 |
| | CodeContests | 165 | 203.7 | 1,000 |

sensus set is the set of all pairs that consist of a code solution from $\mathcal{S}_x$ and a test case from $\mathcal{S}_y$, i.e., $\mathcal{S} = \{(x, y)|x \in \mathcal{S}_x, y \in \mathcal{S}_y\}$. For example in Figure 3, we can get $\mathcal{S}_x = \{x_1, x_2\}, \mathcal{S}_y = \{y_1, y_2, y_3\}$ from the hypothetical inlier $(x_1, y_1)$ (shown in green box), and $\mathcal{S}_x = \{x_3\}, \mathcal{S}_y = \{y_2, y_3, y_4, y_5\}$ from $(x_3, y_2)$ (shown in purple box).

- We score the consensus set as $f(\mathcal{S}) = |\mathcal{S}_x||\mathcal{S}_y|$, where $|\mathcal{S}_x|$ is the number of code solutions in $\mathcal{S}_x$ and $|\mathcal{S}_y|$ is the number of test cases in $\mathcal{S}_y$. This score is equal to the number of pairs in the consensus set. The intuition is that the more pairs that agree with the hypothetical functionality, the more likely this functionality is correct, according to our assumptions. Following the example in Figure 3, the consensus set scores are 6 and 4 for the hypothetical inliers $(x_1, y_1)$ and $(x_3, y_2)$, respectively.

We repeat the above procedure for a fixed number of times, each time producing a consensus set with its score. Finally, we get the best code solution $\hat{x}$ by selecting any code solution from the consensus set with the highest score. If we want to obtain $k$ code solutions, we can select the top $k$ consensus sets with the highest scores, and one code solution is picked up from each of the $k$ consensus sets.

In practice, when the number of code solutions in $\mathcal{D}$ is not large, we can simplify the above method by examining all possible pairs in $\mathcal{D}$, instead of sampling pairs from $\mathcal{D}$. Specially, for each code solution $x \in \mathbf{X}$, we run it with every test case in $\mathbf{Y}$ and keep track of which test cases it passes. We group together code solutions that pass the same test cases, because they have the same functionality. This way, we divide all code solutions in $\mathbf{X}$ into groups based on their functionality, which we write as $\mathbf{X} = \{\mathcal{S}_x^1, \mathcal{S}_x^2, \cdots, \mathcal{S}_x^K\}$, where $K$ is the number of code solution groups. Each group $\mathcal{S}_x$ has a set of test cases that it passes, which we write as $\mathcal{S}_y$. Then, we get $K$ consensus sets, each of which has the form $\mathcal{S} = \{(x, y)|x \in \mathcal{S}_x, y \in \mathcal{S}_y\}$. We can score each consensus set by $f(\mathcal{S}) = |\mathcal{S}_x||\mathcal{S}_y|$, as before. This naive version captures the same underline intuition, but it finds all consensus sets right away, without sampling pairs repeatedly.

## 3 EXPERIMENTAL SETUP

**Models**  Our experiments are based on Codex (Chen et al., 2021), INCODER (Fried et al., 2022) and CODEGEN (Nijkamp et al., 2022). Codex is a descendant of GPT-3 (Brown et al., 2020) and proficient in understanding the provided context and generating functional programs. We use three Codex models with different capabilities provided by OpenAI: code-cushman-001, code-davinci-001, and code-davinci-002. INCODER is a unified generative model that can perform left-to-right code generation and code infilling, while CODEGEN is a family of large-scale language models to perform conversational program synthesis. We take use of the INCODER 6.7B version (INCODER-6B) and the CODEGEN 16B Python mono-lingual version (CODEGEN-MONO-16B).

**Metrics and Baseline**  We use the metric pass@$k$ (with $n$ samples) for performance evaluation and take advantage of ground truth test cases to determine the functional correctness of code solutions. For each problem, we sample $n$ code solutions and then select $k$ of them for evaluation. If any of the $k$ code solutions passes all ground truth test cases, the problem is considered solved. Then pass@$k$ is the percentage of solved problems. We use the unbiased definition of pass@$k$ as our

| Methods | Baseline | | | AlphaCode-C | | | CODET | | |
|---|---|---|---|---|---|---|---|---|---|
| $k$ | 1 | 10 | 100 | 1 | 2 | 10 | 1 | 2 | 10 |
| **HumanEval** | | | | | | | | | |
| code-cushman-001 | 33.5 | 54.3 | 77.4 | 39.6 | 46.4 | 63.8 | 44.5 11.0 | 50.1 | 65.7 11.4 |
| code-davinci-001 | 39.0 | 60.6 | 84.1 | 41.6 | 50.7 | 75.6 | 50.2 11.2 | 58.9 | 75.8 15.2 |
| code-davinci-002 | 47.0 | 74.9 | 92.1 | 55.1 | 64.1 | 84.4 | 65.8 18.8 | 75.1 | 86.6 11.7 |
| INCODER-6B | 16.4 15.2 | 28.3 27.8 | 47.5 47.0 | 17.7 | 23.8 | 34.8 | 20.6 4.2 | 27.6 | 37.1 8.8 |
| CODEGEN-MONO-16B | 29.7 29.3 | 50.3 49.9 | 73.7 75.0 | 27.3 | 38.5 | 64.4 | 36.7 7.0 | 44.7 | 59.3 9.0 |
| **MBPP** | | | | | | | | | |
| code-cushman-001 | 45.9 | 66.9 | 79.9 | 51.5 | 59.0 | 73.3 | 55.4 9.5 | 61.7 | 72.7 5.8 |
| code-davinci-001 | 51.8 | 72.8 | 84.1 | 56.2 | 64.7 | 78.8 | 61.9 10.1 | 69.1 | 79.3 6.5 |
| code-davinci-002 | 58.1 | 76.2 | 84.5 | 62.0 | 70.7 | 79.9 | 67.7 9.6 | 74.6 | 81.5 4.8 |
| INCODER-6B | 21.3 19.4 | 46.5 | 66.2 | 26.7 | 35.3 | 56.2 | 34.4 13.1 | 43.9 | 58.2 11.7 |
| CODEGEN-MONO-16B | 42.4 | 65.8 | 79.1 | 41.0 | 55.9 | 73.6 | 49.5 7.1 | 56.6 | 68.5 2.7 |

Table 2: Pass@$k$ (%) on the HumanEval and MBPP benchmarks. AlphaCode-C is our replication of the clustering method in Li et al. (2022b). The numbers in red indicate the absolute improvements of CODET over baseline on pass@1 and pass@10. We also list the baseline results from Fried et al. (2022) and Nijkamp et al. (2022) for reference in gray, where the settings of context are not exactly the same as ours. For CODET, temperature is set to 0.8 and sampling number is set to 100. We do not show CODET pass@100, since it is the same as the baseline pass@100.

baseline (Chen et al., 2021), where $k$ solutions are randomly picked from $n$ samples. Our CodeT uses a dual execution agreement mechanism to select $k$ solutions from $n$ samples, as mentioned in 2.2. In addition, we include a clustering method from Li et al. (2022b) for comparison, denoted as AlphaCode-C. Our replication is to use the test inputs generated by CODET, run the solutions on the test inputs, group the solutions by test outputs, and rank the clusters by size (details in Appendix I).

**Benchmarks** We conduct experiments on four public code generation benchmarks in the zero-shot setting. The statistics of benchmarks are shown in Table 1. (1) **HumanEval** (Chen et al., 2021) consists of hand-written Python programming problems. The original contexts include example input-output cases, which are removed in our experiments to avoid exposing real test cases. The experiment in Appendix B shows that this removal operation is reasonable and indispensable. (2) **MBPP** (Austin et al., 2021) (sanitized version) contains crowd-sourced Python programming problems, and we follow HumanEval to construct the context for it. (3) **APPS** (Hendrycks et al., 2021) consists of coding problems collected from open-access coding websites, which have different difficulty levels. (4) **CodeContests** (Li et al., 2022b) includes competitive programming problems scraped from the Codeforces platform. To enable zero-shot inference, we construct the context for APPS and CodeContests as follows: the original problem description is treated as a comment where input-output examples are removed, and a simple function header "def solution(stdin : str) $\rightarrow$ str :" is placed after the comment to accommodate the input/output data format. More implementation details can be found in Appendix A.

## 4 EXPERIMENTAL RESULTS

In this section, we evaluate CODET on five different pre-trained models and four benchmarks to verify its effectiveness, followed by test case analysis and case studies to provide more insights.

### 4.1 RESULTS ON HUMANEVAL AND MBPP

The experimental results of various models on the HumanEval and MBPP benchmarks are summarized in Table 2. If we compare the pass@100 to pass@1 on the Baseline column, it is clear that the former is significantly better than the latter, indicating the potential to select the best code solution from the 100 generated samples.

For three Codex models, when we compare the CODET column with the Baseline column, CODET pass@1 achieves an absolute improvement of about 10% over the baseline pass@1. The improvements are consistently above 10% on HumanEval. Surprisingly, even for the strongest baseline,

| Methods | | Baseline | | | | | CODET | | | |
|---|---|---|---|---|---|---|---|---|---|---|
| | $k$ | 1 | 10 | 50 | 100 | 1000 | 1 | 2 | 10 | 100 |
| APPS | INTRODUCTORY | 27.2 | 46.6 | 59.4 | - | - | 34.6 7.4 | 41.2 | 53.2 6.6 | - |
| | INTERVIEW | 5.1 | 12.8 | 23.0 | - | - | 8.1 3.0 | 11.2 | 18.1 5.3 | - |
| | COMPETITION | 1.8 | 4.9 | 12.1 | - | - | 2.2 0.4 | 4.1 | 8.6 3.7 | - |
| CodeContests | | 0.7 | 3.0 | 5.7 | 7.5 | 13.9 | 2.1 1.4 | 2.3 | 5.3 2.3 | 9.9 2.4 |

Table 3: Pass@$k$ (%) results on the APPS and CodeContests benchmarks using code-davinci-002 in the zero-shot setting. The numbers in red indicate the absolute improvements of CODET over baseline on pass@1, pass@10 and pass@100. For CODET, temperature is set to 0.8 and sampling number is set to 50 for APPS and 1,000 for CodeContests.

code-davinci-002, the improvement is 18.8%, boosting the pass@1 to 65.8%, which is a 20+% absolute improvement over the best previously reported results (Inala et al., 2022). We attribute this larger improvement to the higher quality of test cases generated by code-davinci-002, providing a deeper analysis in Section 4.3. CODET also achieves exceptional performance on the MBPP benchmark, although the magnitude of the improvements is slightly less than that of HumanEval. Using the code-davinci-002 as an example, the pass@1 improves by 9.6%. We also report pass@2 and pass@10 of CODET to further show its superiority. The pass@2 results of CODET are close to the baseline pass@10 results. Meanwhile, the improvements on pass@10 are also consistently over 10% on the HumanEval benchmark.

The experimental results of INCODER-6B and CODEGEN-MONO-16B further verify the effectiveness of CODET. It is obvious CODET can significantly improve the pass@1, with absolute improvements in the range of 4.2% to 13.1%. INCODER-6B achieves the greatest improvement with a gain of 13.1% on the MBPP benchmark. Similar to the experimental results of Codex, the pass@2 results are close to the baseline pass@10. All the results demonstrate that CODET can boost the performance of various pre-trained language models consistently.

As for AlphaCode-C, it is consistently inferior to CODET on both benchmarks using different models, demonstrating the superiority of our dual execution agreement that takes test case information into consideration. In addition, we notice that duplication exists in the generated code solutions and test cases. We perform an ablation study in Appendix D to show that de-duplication has little influence on the results of CODET. Moreover, we discuss the sensitivity of CODET to the temperature in Appendix E, showing the rationality of choosing a rather high temperature at 0.8.

## 4.2 RESULTS ON APPS AND CODECONTESTS

We also conduct experiments on two more challenging benchmarks, APPS and CodeContests. We build the zero-shot versions of APPS and CodeContests to be in line with our setting of HumanEval and MBPP by removing the example input-output cases in the problem descriptions. We employ code-davinci-002 for code solution and test case generation. The sampling number is set to 50 for APPS to save computation cost on the 5,000 testing problems, while for CodeContests, following Li et al. (2022b), the sampling number is set to 1,000 to solve especially hard problems. From the results summarized in Table 3, we can clearly observe the consistent performance improvements on both benchmarks using CODET. The absolute pass@1 improvement is 7.4% for introductory problems in APPS, while the improvements are not significant for competition level problems in APPS and CodeContest, indicating their difficulties. In addition, we notice that code-davinci-002 may generate many trivial code solutions for the problems in APPS and CodeContests due to the superior difficulty of these two benchmarks. We perform a comprehensive study in Appendix F to demonstrate the robustness of CODET to this issue. Inspired by Chen et al. (2021) and Li et al. (2022b), we also conduct experiments in the one-shot setting, which is detailed in Appendix G.

## 4.3 ANALYSIS ON TEST CASES

The test cases are vital to CODET since the core idea is based on test-driven execution agreement. Hence, in this subsection, we analyze the test cases by answering the following research questions.

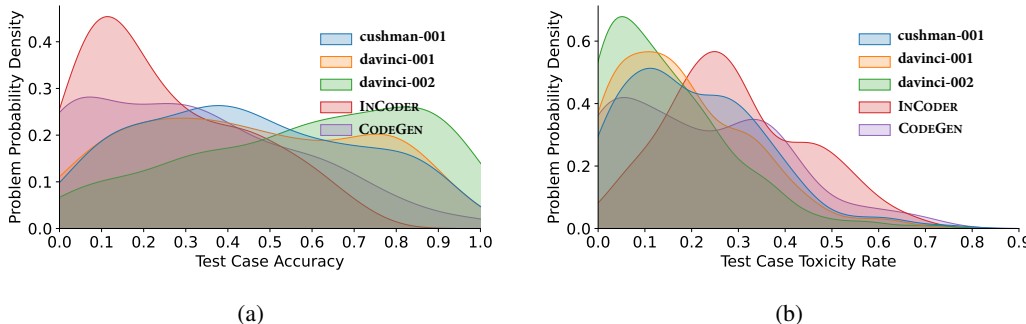

|  | (a) |  | (b) |

Figure 4: The distributions of (a) test case accuracy and (b) toxicity rate for each problem on HumanEval. Test cases are of better quality if they have higher accuracy and lower toxicity rate.

| Benchmarks | HumanEval | | | MBPP | | |
|---|---|---|---|---|---|---|
| $k$ | 1 | 2 | 10 | 1 | 2 | 10 |
| code-cushman-001 | 47.1 2.6 | 58.6 8.5 | 71.2 5.5 | 59.7 4.3 | 64.8 3.1 | 75.5 2.8 |
| code-davinci-001 | 52.0 1.8 | 62.9 4.0 | 78.1 2.3 | 64.3 2.4 | 71.7 2.6 | 80.5 1.2 |
| INCODER-6B | 26.8 6.2 | 30.4 2.8 | 40.8 3.7 | 50.3 15.9 | 55.4 11.5 | 64.5 6.3 |
| CODEGEN-MONO-16B | 47.7 11.0 | 54.9 10.2 | 71.0 11.7 | 60.0 10.5 | 67.6 11.0 | 76.5 8.0 |

Table 4: Pass@$k$ (%) on the HumanEval and MBPP benchmarks with code-cushman-001, code-davinci-001, INCODER, and CODEGEN using the test cases generated by code-davinci-002. The numbers in orange indicate the absolute improvements of pass@$k$ using code-davinci-002 test cases over that using their own generated test cases.

## Q1. What is the quality of the generated test cases?

We evaluate the correctness of the generated test cases using the canonical solutions. A test case is considered correct if the canonical solution can pass it. Figure 4a summarizes the distributions of test case accuracy on HumanEval, where the horizontal axis represents the accuracy value for each problem and the vertical axis represents the probability density of problems with the corresponding accuracy value. We can see that the test cases generated by Codex models are of much higher accuracy than CODEGEN/INCODER. Besides accuracy, we also introduce the test case toxicity rate as a measurement of quality. We consider a test case to be "toxic" if any generated code solution can pass it while the canonical solution cannot. Toxic test cases may hinder the scoring of consensus sets and lead to the failure of CODET. As shown in Figure 4b, we can find that the toxicity rate highly correlates to the test case accuracy with respect to different models, where the proportions of toxic test cases for Codex models are smaller than CODEGEN/INCODER. We also evaluate the code coverage of generated test cases using two coverage criteria in Appendix H.2, where Codex models still outperform CODEGEN/INCODER with an average coverage of over 95%. Comparing the test case quality and the performance of CODET shown in Table 2, we can find that the quality of test cases strongly correlates to the performance gain using CODET concerning different models.

## Q2. Can better test cases further boost the performance of mediocre models?

From the above discussion with Figure 4, we can find that code-davinci-002 is the most capable model for generating high-quality test cases. Hence, we conduct an experiment to boost the performance of the other four models (code-cushman-001, code-davinci-001, INCODER, and CODEGEN) using test cases generated by code-davinci-002. Table 4 summarizes the performance gain with respect to different models on the HumanEval and MBPP benchmarks. In general, using the test cases generated by code-davinci-002 has significantly better performance than using the test cases generated by the less capable models themselves. For code-cushman-001 and code-davinci-001, the absolute improvements are in the range of 1.8% to 4.3% on pass@1, while for INCODER and CODEGEN, the range is from 6.2% to 15.9%. The above results indicate that the correct code solutions generated by mediocre models can be further exploited by adopting better test cases.

```python
def below_threshold(l: list, t: int):
    """
    Return True if all numbers in the
    list l are below threshold t.
    """
```

Rank #1: The consensus set has **61** solutions and **218** test cases.  **Correct**

```python
    if l == []: return True
    return l[0] < t and
    below_threshold(l[1:], t)
```

Rank #2: The consensus set has **30** solutions and **226** test cases.  **Incorrect**

```python
    for e in l:
        if e > t: return False
    return True
```

(a)

```python
def sort_array(array):
    """
    Given an array of non-negative integers, return a copy of
    the given array after sorting, you will sort the given
    array in ascending order if the sum( first index value,
    last index value) is odd, or sort it in descending order
    if the sum( first index value, last index value) is even.
    """
```

Rank #1: The consensus set has **4** solutions and **138** test cases.  **Incorrect**

```python
    sum = 0
    for i in range(len(array)):
        sum += array[i]
    if sum % 2 == 0:
        array.sort(reverse=True)
    else: array.sort()
    return array
```

Rank #2: The consensus set has **3** solutions and **158** test cases.  **Correct**

```python
    initial_sum = sum(array[0:1])
    + sum(array[-1:])
    if initial_sum % 2 == 0:
        return sorted(array,
        reverse=True)
    else: return sorted(array)
```

(b)

Figure 5: Two real cases from the HumanEval benchmark with CODET and code-cushman-001.

## Q3. How effective is CODET when there are fewer test cases?

|        | **Sampling Number** | | | |
|--------|------|------|------|------|
| *Limit* | 10 | 20 | 50 | 100 |
| 1 | 56.5 | 57.5 | 60.7 | 62.4 |
| 2 | 62.2 | 62.8 | 63.2 | 63.6 |
| 3 | 62.9 | 63.2 | 65.5 | 65.0 |
| 4 | 64.1 | 64.5 | 65.7 | 65.0 |
| 5 | 63.9 | 64.2 | 65.2 | 65.8 |

Table 5: Pass@1 (%) on HumanEval using CODET and code-davinci-002 with different numbers of test cases. *Sampling Number* denotes the number of samples generated by model, and *Limit* denotes the test cases extracted per sample.

When generating test cases for the HumanEval benchmark, we sample 100 times for each problem and each sample may include multiple assertion statements (i.e., test cases), denoted as *Sampling Number* = 100. Then we extract the first 5 syntactically correct test cases from each sample, denoted as *Limit* = 5. This means each problem is equipped with 500 test cases at most. The actual numbers of extracted test cases are summarized in Appendix H.1. We perform an ablation study on the number of test cases by decreasing *Sampling Number* and *Limit*. As shown in Table 5, we can conclude that using more test cases in CODET could generally lead to better performance, while the performance gap narrows when *Sampling Number* $\geq$ 50 and *Limit* $\geq$ 3. Moreover, CODET improves the pass@1 by 9.5% with only 10 test cases using code-davinci-002, suggesting the high test case efficiency. We can use a smaller *Sampling Number* in real-world application to balance the performance and computation cost. More results can be found in Appendix H.3.

## 4.4 CASE STUDY

In CODET, we design the dual execution agreement based on the idea that a good code solution can pass the most test cases and agree with the most solutions of the same functionality. We use "dual" because both the code solutions and the test cases are critical. Figure 5a shows a case from the HumanEval benchmark using code-cushman-001. The highest scoring consensus set has the correct functionality that returns true if all numbers in the list are below threshold $t$, while the consensus set ranked 2 does not understand the boundary condition exactly. The solutions in the second consensus set can pass more test cases (i.e., 226) than that in the first consensus set (i.e., 218). However, considering both code solutions and test cases, CODET can successfully rank the consensus sets and find the correct solutions. Such cases are not rare, suggesting that our design of the dual execution agreement is reasonable. For further statistical demonstration, we conduct an ablation study to score the consensus set by considering only the number of code solutions or test cases. The results again support our claim, as detailed in Appendix I.

CODET is empowered by the pre-trained language models, but is also limited by them. Therefore, the second assumption made in Section 2.2 does not always hold, leading to error cases where the correct code solution is generated, but not in the top 1 consensus set. For CODET with code-cushman-001 on the HumanEval benchmark, we find 53 out of 164 programming problems that belong to this situation. We manually investigated these problems and found that 20% of them can be blamed on issues such as ambiguous problem descriptions, uncovered corner cases, and lack

of import statements, while the remaining problems are attributed to the failure of the model to understand the problem descriptions. Figure 5b shows an error case caused by ambiguity. The correct understanding of the description "sum(first index value, last index value)" is to add the first and last values, while the code solutions that sum all values from the first to the last are ranked top 1. More real cases can be found in Appendix J. And hope the error analysis can provide inspiration for future studies on improving code generation for more difficult programming problems.

## 5 RELATED WORK

**Code Generation with Large Models** Recently, a number of large pre-trained language models have been proposed for code generation. Benefiting from billions of trainable parameters and massive publicly available source code, models could achieve surprisingly good performance. For instance, AlphaCode (Li et al., 2022b) claimed to have outperformed half of the human competitors in real-world programming competitions, and Codex (Chen et al., 2021) is empowering Copilot to provide real-time coding suggestions. Other open-source code generation models include GPT-Neo (Black et al., 2021), GPT-J (Wang & Komatsuzaki, 2021), CodeParrot (Tunstall et al., 2022), PolyCoder (Xu et al., 2022), CODEGEN (Nijkamp et al., 2022), and INCODER (Fried et al., 2022). In our study, we take advantage of the Codex inference API provided by OpenAI as well as the two competitive open-source models CODEGEN and INCODER to perform zero-shot code generation.

**Automatic Test Case Generation** Automated test case generation for programming problems can reduce the effort of writing test cases manually by developers. Early works including Randoop (Pacheco et al., 2007), EvoSuite (Fraser & Arcuri, 2011), MOSA (Panichella et al., 2015), DynaMOSA (Panichella et al., 2017), and MIO (Arcuri, 2017), were proposed to automatically generate test cases for statically typed programming languages like Java. The later proposed Pynguin (Lukasczyk & Fraser, 2022) could handle dynamically typed language like Python. Nevertheless, they are all search-based heuristics methods, which have limitations to the diversity and quantity of generated test cases. To combat these limitations, recently proposed approaches (Tufano et al., 2020; Li et al., 2022b) leveraged pre-trained language models like BART (Lewis et al., 2019) and T5 (Raffel et al., 2020) fine-tuned on labelled data for test case generation. Unlike previous works that require heuristic rules or model training, we directly sample test cases from powerful code generation models like Codex in the zero-shot setting with elaborate prompts.

**Code Selection from Multiple Samples** Despite large models have achieved great performance in code generation, the models need to sample many times to find the correct answer. Recently, several approaches were proposed to tackle this issue. In the domain of solving math word problems, Cobbe et al. (2021) chose the one with highest rank by a trained verifier, and Shen et al. (2021) proposed to jointly train the generator and ranker through a multi-task framework. In the domain of general purpose code generation, Inala et al. (2022) trained a fault-aware ranker. Moreover, some work has been proposed to leverage the execution information (Shi et al., 2022; Li et al., 2022b; Le et al., 2022; Lahiri et al., 2022). Unlike previous works that require model training or pre-existing test cases or user interactions, we let the large models generate test cases for themselves and automatically rank the solutions based on the test-driven dual execution agreement. The idea of ranking based on agreement also appears in the domain of reasoning (Wang et al., 2022; Li et al., 2022a).

## 6 CONCLUSION AND FUTURE WORK

In this paper, we propose a simple yet effective approach, called CODET, leveraging pre-trained language models to generate both the code solutions and the test cases. CODET executes the code solutions using the test cases and chooses the best solution based on the dual execution agreement. We demonstrate the dual agreement with both the test cases and other solutions is critical to the success of CODET, perform a thorough analysis on the quality of generated test cases and their impact on CODET, and study cases to provide more insights. Experimental results clearly demonstrate the superiority of CODET, improving the pass@1 numbers significantly on various benchmarks. While there remain challenges that CODET only works for executable code generation and it introduces extra computation cost for test case generation. In future work, we will explore the ways to tackle these challenges and improve CODET to solve more difficult programming problems.

ACKNOWLEDGEMENT

We would like to thank Davis Mueller and Jade Huang for proofreading the paper and providing valuable comments. We also sincerely thank all the anonymous reviewers for their constructive feedback and insightful comments.

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

| Methods | Baseline | | | CODET | | |
|---|---|---|---|---|---|---|
| $k$ | 1 | 10 | 100 | 1 | 2 | 10 |
| code-cushman-001 | 31.7 $_{-1.8}$ | 56.4 $_{2.1}$ | 84.1 $_{6.7}$ | 58.6 $_{14.1}$ | 65.7 $_{15.6}$ | 80.1 $_{14.4}$ |
| code-davinci-001 | 34.8 $_{-4.2}$ | 63.0 $_{2.4}$ | 87.2 $_{3.1}$ | 60.4 $_{10.2}$ | 69.1 $_{10.2}$ | 82.4 $_{6.6}$ |
| code-davinci-002 | 47.6 $_{0.6}$ | 78.8 $_{3.9}$ | 92.7 $_{0.6}$ | 74.8 $_{9.0}$ | 82.9 $_{7.8}$ | 89.0 $_{2.4}$ |

Table 6: Pass@$k$ (%) on the original HumanEval benchmark with Codex models. The numbers in orange indicate the absolute improvements of pass@$k$ on the original benchmark over our modified benchmark in Table 2.

## A  MORE IMPLEMENTATION DETAILS

We set the temperature to $0.8$, the top $p$ to $0.95$, the max generation length to $300$, and the timeout of executing a test case to $0.1$ seconds. Specially, for baseline pass@1, we use the greedy search setting with temperature 0. The number of sampling test cases for each problem is set to 100 for the HumanEval and MBPP benchmarks, and 50 for the APPS and CodeContests benchmarks. When scoring consensus sets in CODET, we use the square root of $|\mathcal{S}_x|$ to reduce the impact caused by code solutions. A supporting experiment can be found in Appendix C. For code solution post-processing, we follow Chen et al. (2021) to truncate the generated content by five stop sequences: "\nclass", "\ndef", "\n#", "\nif", and "\nprint". For the implementation of INCODER and CODEGEN, we use the HuggingFace transformers library (Wolf et al., 2019) and run both models with half precision. In addition, when the number of consensus sets in CODET is smaller than $k$, the selection is done from the highest scoring consensus set to the lowest. When reaching the set with the lowest score, it repeats from the highest scoring consensus set. In most cases, the number of consensus sets is larger than $k$, as shown in Figure 6.

## B  RESULTS ON ORIGINAL HUMANEVAL

As mentioned in Section 3, for all benchmarks, we remove the example input-output cases from the original contexts to avoid exposing real test cases. To study the influence of such modification, we take HumanEval as an example and perform an additional experiment with its original contexts. The results are summarized in Table 6. On the one hand, the baseline pass@10 and pass@100 results on the original HumanEval benchmark outperform the modified version, which is reasonable because the example input-output cases may provide useful information for code generation. Nevertheless, the pass@1 results on the original benchmark are basically the same or even worse than the modified version, suggesting that the Codex models have not fully understood the semantics of the example input-output cases provided in the contexts. On the other hand, the performance of CODET is significantly improved using the original benchmark. This is as expected because the original contexts used for test case generation include real test cases, which could be borrowed by the models during the generation. Such real test cases will greatly empower CODET to distinguish correct code solutions. Hence, in our experiments, it is indispensable to remove the example input-output cases to avoid exposing the real test cases. In this way, the effectiveness of CODET can be fairly verified.

## C  ANALYSIS ON CODE SOLUTIONS

In CODET, code solutions that can pass exactly the same test cases are considered consistent in functionality and are grouped into the same consensus set. Since we employ top $p$ sampling with a rather high temperature of $0.8$, the functionality of the code solutions may vary significantly, which results in more consensus sets. We draw a histogram in Figure 6 to show the number of consensus sets produced by code-cushman-001 and CODET for each problem on the HumanEval benchmark. The average and median numbers are $26.8$ and $25.5$, respectively. We can find that most problems have less than 50 consensus sets, but the numbers have a high variance among different problems. We also draw the distribution of the numbers of code solutions for the top-ranked consensus sets in Figure 7. The consensus sets ranked top 1 tend to have more code solutions with an average value of $9.8$, and the numbers also have a high variance.

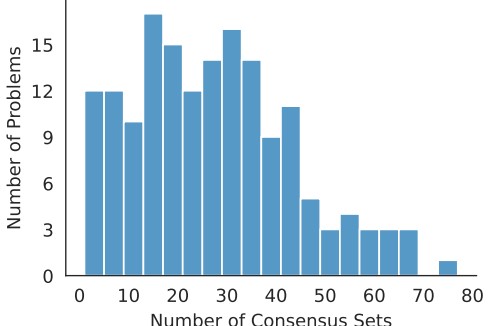

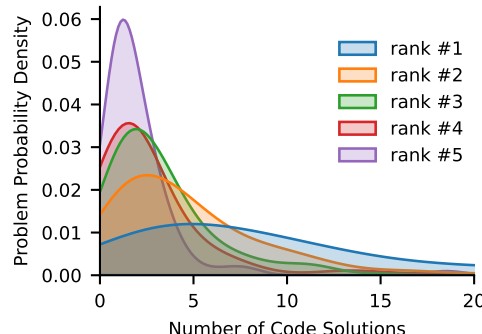

Figure 6: The numbers of consensus sets that are produced by code-cushman-001 and CODET on the HumanEval benchmark.

Figure 7: The distribution of the code solution numbers for the top 5 consensus sets. The long tail distribution with number $\geq 20$ is truncated.

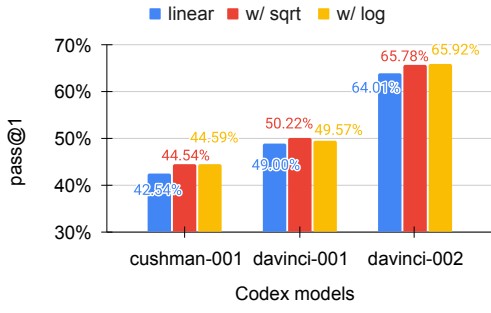

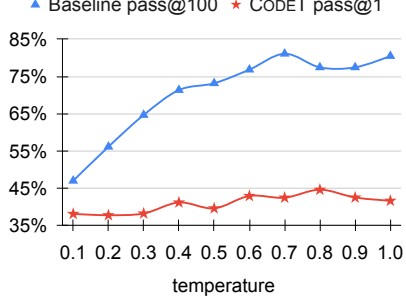

Figure 8: The CODET results of three Codex models with and without constraint on the number of code solutions.

Figure 9: The baseline pass@100 and CODET pass@1 with code-cushman-001 at different temperature settings.

As mentioned in Appendix A, we use the square root of $|\mathcal{S}_x|$ to reduce the impact caused by code solutions, because we believe passing more test cases is more important than having more code solutions with the same functionality. For example, there may be one code solution that can pass five test cases, whereas another five code solutions in a consensus set can pass only one test case. We intuitively consider that the former may be more likely correct. For validation, we perform an experiment by comparing the performance of CODET with the "sqrt", "log" functions, and without any constraint (i.e., "linear") on the number of code solutions. Figure 8 shows the results of three Codex models on the HumanEval benchmark. We can find that reducing the importance of code solutions can consistently improve the performance of CODET. Similar observations have been found in other models and benchmarks, where the performance of employing "sqrt" is always better than or competitive to "linear", indicating the rationality of our design.

## D   INFLUENCE OF DE-DUPLICATION

Since we sample multiple times during generation, there is the chance that many of the generated code solutions and test cases are exactly the same. On the one hand, the number of duplicates may indicate the importance of a sample. On the other hand, duplicates may hinder the scoring of consensus sets in CODET when the quality of generation is unsatisfactory. Hence, we perform an ablation study to investigate the effects of removing duplicate code solutions and test cases. Specifically, we first format the generated Python code to conform to the PEP 8 style guide[3], and then remove duplicate code solutions and test cases before performing CODET. The de-duplication results on the HumanEval and MBPP benchmarks using CODET and code-cushman-001 are shown in Table 7, where we can choose to de-duplicate the code solutions, or the test cases, or both. We can

---

[3] https://peps.python.org/pep-0008

| De-duplication | | HumanEval | | | MBPP | | |
|---|---|---|---|---|---|---|---|
| Solution | Test | 1 | 2 | 10 | 1 | 2 | 10 |
| No | No | 44.5 | 50.1 | 65.7 | **55.4** | 61.7 | 72.7 |
| No | Yes | 42.2 | 48.8 | **66.7** | 54.5 | **62.3** | **73.4** |
| Yes | No | **46.9** | **52.5** | 65.6 | 54.7 | 61.7 | 73.2 |
| Yes | Yes | 42.7 | 51.2 | 66.4 | 54.7 | 62.1 | 73.2 |

Table 7: Pass@$k$ (%) on the HumanEval and MBPP benchmarks using CODET and code-cushman-001 with different de-duplication settings. The setting "No No" in the first line means that neither the code solutions nor the test cases are de-duplicated, which is used in our main experiments.

| Methods | | CODET | | | CODET (Remove Trivial) | | |
|---|---|---|---|---|---|---|---|
| $k$ | | 1 | 10 | 100 | 1 | 10 | 100 |
| APPS | INTRODUCTORY | 34.6 | 53.2 | - | 34.9 0.3 | 53.4 0.2 | - |
| | INTERVIEW | 8.1 | 18.1 | - | 8.3 0.2 | 18.2 0.1 | - |
| | COMPETITION | 2.2 | 8.6 | - | 2.5 0.3 | 8.7 0.1 | - |
| CodeContests | | 2.1 | 5.3 | 9.9 | 2.7 0.6 | 5.3 0.0 | 10.0 0.1 |

Table 8: Pass@$k$ (%) results on the zero-shot APPS and CodeContests benchmarks using code-davinci-002 and CODET with/without the trivial code solutions filtered. The numbers in red indicate the absolute improvements after filtering the trivial solutions.

find that de-duplication has slight and inconsistent influence on the performance of CODET. For the HumanEval benchmark, the pass@1 results using code solution de-duplication alone are better than other settings. Nonetheless, for the MBPP benchmark, the best pass@1 results are achieved without de-duplication. Therefore, in our main experiments, we reserve all the generated code solutions and test cases when performing CODET and leave the study of more advanced de-duplication methods for future work.

## E  SENSITIVITY TO THE TEMPERATURE

The hyper-parameter temperature has a great impact on the quality of generated code solutions and test cases when using top $p$ sampling. We use a high temperature of $0.8$ in our main experiments since CODET could benefit from a larger number of diverse samples. To investigate the sensitivity of CODET to the temperature, we perform an ablation study by using a range of temperatures to report the results of baseline pass@100 and CODET pass@1. Figure 9 shows the results of code-cushman-001 on the HumanEval benchmark at different temperature settings. We can find that a higher temperature does improve the baseline pass@100 and CODET pass@1, and CODET achieves a good performance when temperature is set to $0.8$.

## F  REMOVING TRIVIAL CODE SOLUTIONS

The problems in the APPS COMPETITION and CodeContests benchmarks are of great difficulty compared to HumanEval and MBPP, leading to the poor performance of the most capable code-davinci-002 model. After checking the incorrect code solutions generated by code-davinci-002, we identify many trivial solutions that just return the input argument or a constant value. Such solutions may hinder the ranking process of CODET if they can pass any generated test case. A trivial solution can be easily identified by its input arguments and returned values. If a solution always returns the same output value for different inputs, or its returned values are always the same as the inputs, it must be a trivial solution. To investigate the impact of trivial code solutions, we use code-davinci-002 on the zero-shot APPS and CodeContests benchmarks, and perform CODET after filtering out all the trivial solutions. As a result, we can remove an average of $4.5$ ($91.6$) trivial solutions from the $50$ ($1,000$) generated solutions per problem for the APPS (CodeContests) benchmark. How-

| | $k$ | 1 | 10 | 50 | 100 | 1000 | 1 | 2 | 10 | 100 |
|---|---|---|---|---|---|---|---|---|---|---|
| | | **Baseline** | | | | | **CODET** | | | |
| APPS | INTRODUCTORY | 29.3 | 48.5 | 60.9 | - | - | 47.3 18.0 | 52.7 | 58.4 9.9 | - |
| | INTERVIEW | 6.4 | 14.6 | 25.4 | - | - | 14.3 7.9 | 18.2 | 23.3 8.7 | - |
| | COMPETITION | 2.5 | 6.3 | 14.5 | - | - | 6.2 3.7 | 9.8 | 13.6 7.3 | - |
| CodeContests | | 1.0 | 4.1 | 7.1 | 8.8 | 15.2 | 3.2 2.2 | 5.6 | 9.3 5.2 | 12.3 3.5 |
| | | **Baseline Filter** | | | | | **CODET Filter** | | | |
| APPS | INTRODUCTORY | 43.6 | 58.6 | - | - | - | 49.6 6.0 | 54.3 | 59.4 0.8 | - |
| | INTERVIEW | 13.3 | 22.8 | - | - | - | 16.1 2.8 | 19.5 | 24.0 1.2 | - |
| | COMPETITION | 7.0 | 13.3 | - | - | - | 7.9 0.9 | 10.5 | 14.1 0.8 | - |
| CodeContests | | 9.9 | 14.5 | 15.1 | 15.2 | - | 9.6 −0.3 | 11.5 | 13.7 −0.8 | 14.5 −0.7 |

Table 9: Pass@$k$ (%) results on the APPS and CodeContests benchmarks using code-davinci-002 and the one-shot setting. The numbers in red indicate the absolute improvements of CODET (Filter) over Baseline (Filter) on pass@1, pass@10 and pass@100. For CODET (Filter), temperature is set to 0.8 and sampling number is set to 50 for APPS and 1,000 for CodeContests. We do not report pass@1000 for "Baseline Filter" because the numbers of code solutions after filtering are less than the sampling numbers.

ever, as shown in Table 8, after removing a prominent percentage of trivial solutions, there is little performance gain, which could exactly demonstrate the robustness of CODET.

## G  RESULTS ON APPS AND CODECONTESTS IN THE ONE-SHOT SETTING

Inspired by Chen et al. (2021) and Li et al. (2022b), we build one-shot versions of APPS and Code-Contests by appending a single input-output example to the problem description as a formatting hint. After generation, we filter out the generated solutions that cannot pass the given example input-output cases, which we call the "Baseline Filter" method. After filtering, we can still perform CODET using the rest of code solutions, called the "CODET Filter" method. Following the zero-shot experiments on APPS and CodeContests, we employ code-davinci-002 for generation and set the sampling number to 50 for APPS and 1,000 for CodeContests.

We summarize the experimental results in Table 9, where we can find the one-shot performance using CODET is much better than that reported in Table 3 in the zero-shot setting. The performance of the baselines can be significantly improved by filtering the solutions with the given example test cases. Moreover, "CODET Filter" can further outperform "Baseline Filter" on the APPS benchmark, especially for the introductory and interview problems. Nonetheless, for CodeContests and the competition level problems in APPS, "CODET Filter" has little performance improvement or even performs slightly worse than "Baseline Filter". After manual investigation, we blame such issue to the generated low-quality test cases, which hinder the scoring of consensus sets. This suggests the interest of future study on test case generation for more challenging programming problems.

## H  MORE ANALYSIS ON TEST CASES

### H.1  STATISTICS ON TEST CASES

How many valid test cases do the models generate for CODET? Taking the HumanEval benchmark as an example, we sample 100 times for each problem when generating test cases. As illustrated in Figure 2, at each time of sampling, we feed the *context c* along with an *instruction p* to the model and get the generated content that may contain multiple test cases. Then, as mentioned in Section 4.3, we further post-process the generated samples to get individual test cases that are syntactically correct. Finally, we only keep the first five valid test cases for each sample, which means a problem can be equipped with 500 test cases at most. Table 10 summarizes the average and median numbers of the extracted test cases for each problem. We can find that almost all the models could generate a considerable number of syntactically correct test cases, while CODEGEN generates plenty of unexpected noise.

| Methods | Test Case Number | |
|---|---|---|
| | Average | Median |
| code-cushman-001 | 410.7 | 429.0 |
| code-davinci-001 | 381.9 | 388.0 |
| code-davinci-002 | 391.1 | 402.0 |
| INCODER | 390.1 | 400.0 |
| CODEGEN | 55.6 | 42.0 |

| Methods | Code Coverage | |
|---|---|---|
| | Statement | Branch |
| code-cushman-001 | 95.3 | 98.1 |
| code-davinci-001 | 94.9 | 97.6 |
| code-davinci-002 | 95.7 | 98.5 |
| INCODER | 94.0 | 96.3 |
| CODEGEN | 78.2 | 78.6 |

Table 10: The numbers of extracted test cases for each problem generated by five models on the HumanEval benchmark.

Table 11: The Code Coverage (%) statistics of test cases generated by five models on the HumanEval benchmark.

| Limit | Sampling Number | | | |
|---|---|---|---|---|
| | 10 | 20 | 50 | 100 |
| code-cushman-001 | | | | |
| 1 | 37.8 | 40.0 | 40.8 | 38.7 |
| 2 | 42.1 | 41.8 | 43.4 | 41.8 |
| 3 | 41.6 | 41.9 | 43.8 | 42.5 |
| 4 | 41.2 | 41.2 | 43.8 | 43.3 |
| 5 | 41.0 | 41.9 | 45.4 | 44.5 |
| code-davinci-002 | | | | |
| 1 | 56.5 | 57.5 | 60.7 | 62.4 |
| 2 | 62.2 | 62.8 | 63.2 | 63.6 |
| 3 | 62.9 | 63.2 | 65.5 | 65.0 |
| 4 | 64.1 | 64.5 | 65.7 | 65.0 |
| 5 | 63.9 | 64.2 | 65.2 | 65.8 |

(a) pass@1

| Limit | Sampling Number | | | |
|---|---|---|---|---|
| | 10 | 20 | 50 | 100 |
| code-cushman-001 | | | | |
| 1 | 43.3 | 48.1 | 48.2 | 49.1 |
| 2 | 48.1 | 48.1 | 49.5 | 49.8 |
| 3 | 49.0 | 47.7 | 48.7 | 48.7 |
| 4 | 49.2 | 47.9 | 49.4 | 49.1 |
| 5 | 48.3 | 48.5 | 48.9 | 50.1 |
| code-davinci-002 | | | | |
| 1 | 65.1 | 67.8 | 71.9 | 71.5 |
| 2 | 71.7 | 73.2 | 74.2 | 74.1 |
| 3 | 73.2 | 73.5 | 75.1 | 75.0 |
| 4 | 73.3 | 74.1 | 75.5 | 74.3 |
| 5 | 73.5 | 74.3 | 74.5 | 75.1 |

(b) pass@2

| Limit | Sampling Number | | | |
|---|---|---|---|---|
| | 10 | 20 | 50 | 100 |
| code-cushman-001 | | | | |
| 1 | 55.1 | 56.6 | 61.9 | 62.9 |
| 2 | 58.7 | 61.4 | 64.5 | 65.8 |
| 3 | 60.9 | 62.5 | 63.4 | 65.3 |
| 4 | 61.4 | 63.3 | 63.3 | 65.8 |
| 5 | 63.1 | 62.6 | 63.8 | 65.7 |
| code-davinci-002 | | | | |
| 1 | 77.9 | 79.6 | 82.8 | 84.3 |
| 2 | 80.8 | 81.8 | 84.3 | 86.5 |
| 3 | 82.3 | 83.2 | 85.5 | 87.1 |
| 4 | 82.9 | 84.4 | 85.4 | 86.9 |
| 5 | 83.8 | 84.1 | 85.2 | 86.6 |

(c) pass@10

Table 12: Pass@$k$ (%) on the HumanEval benchmark using CODET with different test case numbers. *Sampling Number* is the number of test case samples we generate for each problem. Each sample may contain multiple assertion statements. These assertion statements are potential test cases, but we do not use all of them. Instead, we extract a *Limit* number of syntactically correct assertion statements from each sample, and discard the rest.

## H.2 CODE COVERAGE OF TEST CASES

To further inspect the quality of generated test cases, we utilize the code coverage measurement and report two coverage criteria — the statement coverage and the branch coverage. The statement coverage can be calculated as the percentage of statements in a code solution that are executed by test cases. The branch coverage is the percentage of executed branches for the control structure (e.g. the if statement). We execute the canonical solution for each HumanEval problem on the test cases generated by five models, then collect the coverage results using Coverage.py[4]. As a result, the average numbers of statements and branches in the canonical solution of a problem are 6.30 and 4.42, respectively. As shown in Table 11, all the models except CODEGEN have good performance on both statement and branch coverage, reaching an average of over 94% coverage. Such results may be attributed to the relatively short canonical solutions and the massive sampling number of test cases. Nevertheless, there are still corner cases that the models cannot cover, which calls for future improvements.

## H.3 RESULTS OF REDUCING THE NUMBER OF TEST CASES

To investigate the performance of CODET using fewer test cases, we perform an ablation study on the number of test cases that participate in the dual execution agreement. As shown in Table 12, we report the results on the HumanEval benchmark using code-cushman-001 and code-davinci-002 with a range of test case numbers. The number of test cases is related to two hyper-parameters. One is the number of test case samples, which is set to 100 for HumanEval in our main experiments. The

---

[4]https://coverage.readthedocs.io/en/6.4.2

| Methods | Code Solution Only $f'$ | | | Test Case Only $f''$ | | |
|---|---|---|---|---|---|---|
| $k$ | 1 | 2 | 10 | 1 | 2 | 10 |
| code-cushman-001 | $41.2^{-3.3}_{+7.7}$ | $49.2^{-0.9}$ | $61.9^{-3.8}_{+7.6}$ | $29.9^{-14.6}_{-3.6}$ | $36.6^{-13.5}$ | $59.5^{-6.2}_{+5.2}$ |
| code-davinci-001 | $44.4^{-5.8}_{+5.4}$ | $54.7^{-4.2}$ | $69.0^{-6.8}_{+8.4}$ | $35.0^{-15.2}_{-4.0}$ | $46.0^{-12.9}$ | $70.2^{-5.6}_{+9.6}$ |
| code-davinci-002 | $55.9^{-9.9}_{+8.9}$ | $67.0^{-8.1}$ | $82.7^{-3.9}_{+7.8}$ | $58.4^{-7.4}_{+11.4}$ | $65.1^{-10.0}$ | $86.1^{-0.5}_{+11.2}$ |

Table 13: Pass@$k$ (%) on the HumanEval benchmark with ranking only on the number of code solutions ($f'(\mathcal{S}) = |\mathcal{S}_x|$) or test cases ($f''(\mathcal{S}) = |\mathcal{S}_y|$) in a consensus set. The numbers in red and green indicate the absolute improvements over baseline and CODET, respectively.

other one is *Limit* that controls the amount of syntactically correct test cases we extract from each sample, which is set to 5 for all benchmarks in our main experiments. Note that *Limit* multiplied by the *Sampling Number* is the maximum number of test cases for a problem, not the exact number, because not every sample contains the *Limit* number of valid test cases. A valid test case (i.e., assertion statement) should start with "assert" and contain the name of the corresponding entry point function. We can conclude from the results that using more test cases in CODET could generally lead to better performance. While the performance gap narrows when *Limit* $\geq 3$ and the sampling number $\geq 50$. Moreover, using only 10 test cases per problem for CODET can still improve the baseline pass@1 performance of code-cushman-001 by absolute 4.3% and code-davinci-002 by absolute 9.5%. It demonstrates that CODET has high test case efficiency and we can use a smaller *Sampling Number* in real-world application to balance the performance and computation cost.

# I   ABLATION STUDY ON THE SCORE OF CONSENSUS SET

In CODET, the score of a consensus set is calculated as $f(\mathcal{S}) = |\mathcal{S}_x||\mathcal{S}_y|$, where $\mathcal{S}_x$ and $\mathcal{S}_y$ are the code solutions and test cases in the consensus set, respectively. We can naturally derive two variants of scoring. One is $f'(\mathcal{S}) = |\mathcal{S}_x|$, in line with the idea of self-consistency (Wang et al., 2022), which only considers the number of code solutions with the same functionality. The other one is $f''(\mathcal{S}) = |\mathcal{S}_y|$, which corresponds to simply counting the test cases that each code solution can pass. To evaluate the performance of these two variants, we perform an ablation study on the HumanEval benchmark using three Codex models. The experimental results are summarized in Table 13, from which we can observe that only considering the number of code solutions or test cases for consensus set scoring performs consistently worse than CODET, and even worse than the baseline. Therefore, it is essential to consider the importance of both code solutions and test cases, suggesting the reasonable design of our dual execution agreement.

As mentioned in Section 3, AlphaCode (Li et al., 2022b) also includes a clustering method (denoted as AlphaCode-C) to select the generated code solutions, which shares a similar goal with our ablation method $f'$: clustering code solutions based on code functionality, and then scoring each cluster by size. AlphaCode-C requires a number of additional test inputs to produce outputs from code solutions, which are then used to determine the functional equivalence. AlphaCode-C relies on a separate test input generation model, which needs extra training and annotation. The model is unavailable and hard to replicate, as the paper does not provide sufficient details. We replicate AlphaCode-C by extracting test inputs from the test cases generated by CODET. We run all code solutions on the test inputs, and group them by outputs. The clusters are ranked by size and then we select the code solutions from each cluster in order. From Table 2 and Table 13, we can find that AlphaCode-C is inferior to $f'$, though they share the similar idea. The reason is that AlphaCode-C will group the trivial code solutions (e.g., solutions that always output "None", "0", or an empty string with whatever inputs) together, leading to a large cluster of incorrect solutions that significantly affects performance. While such trivial code solutions are hard to pass the generated test cases in CODET, thus having lower consensus scores for ranking. This confirms the effectiveness of considering test case information.

```
def remove_vowels(text):
    """

    remove_vowels is a function that takes string and returns string without vowels.
    """
```

| Rank #1: The consensus set has **31** solutions and **170** test cases. An example solution:                    **Correct** | Rank #2: The consensus set has **41** solutions and **147** test cases. An example solution:                  **Incorrect** |
|---|---|
| ```python
    vowels = 'aeiouAEIOU'
    text_without_vowels = ''
    for character in text:
        if character not in vowels:
            text_without_vowels += character
    return text_without_vowels
``` | ```python
    vowels = set("aeiou")
    characters = set(text)
    if not (characters - vowels):
        return ""
    else:
        return "".join(c for c in text
                       if c not in vowels)
``` |

(a) The first consensus set has fewer code solutions.

```
def below_zero(operations: List[int]) -> bool:
    """
    You're given a list of deposit and withdrawal operations on a bank account that starts with
    zero balance. Your task is to detect if at any point the balance of account falls below
    zero, and at that point function should return True. Otherwise it should return False.
    """
```

| Rank #1: The consensus set has **79** solutions and **248** test cases. An example solution:                    **Correct** | Rank #2: The consensus set has **6** solutions and **255** test cases. An example solution:                  **Incorrect** |
|---|---|
| ```python
    zero_balance = 0
    for operation in operations:
        zero_balance += operation
        if zero_balance < 0:
            return True
    return False
``` | ```python
    return any(i < 0 for i in operations)
``` |

(b) The first consensus set has fewer test cases.

Figure 10: Two cases from the HumanEval benchmark, where CODET can find the correct consensus sets though they have (a) fewer code solutions, or (b) fewer test cases.

## J    MORE EXAMPLES FOR CASE STUDY

Figure 10 illustrates two cases that CODET can successfully find the correct consensus sets. Specifically, the case in Figure 10a requires to remove the vowels in the input text. There are $41$ incorrect solutions and $147$ test cases in the consensus set ranked $2$, which forget to remove the upper-case vowels. Though the correct solutions in the top $1$ consensus set are fewer (i.e., $31$), they can pass more test cases (i.e., $170$) and thus have a higher score. The case in Figure 10b is to decide when the balance of account will fall below zero. The functionality of the incorrect solutions in the second consensus set is to tell whether there are withdrawing operations. Nevertheless, the incorrect solutions can pass more test cases (i.e., $255$) than the correct solutions (i.e., $248$) in the top $1$ consensus set. Fortunately, there are $79$ correct solutions and only $6$ incorrect solutions, making it possible for CODET to rank the correct consensus ahead. Both cases demonstrate the plausibility of using the dual execution agreement instead of solely considering the functional agreement between code solutions or the number of passed test cases.

Figure 11 illustrates the cases that CODET fails to find the correct consensus sets. Specifically, Figure 11a demonstrates the situation that there are partially correct solutions that may fail at certain corner cases. In the example, there are $20$ incorrect solutions in the top $1$ consensus set that can pass $205$ test cases, which will fail if the input is a string of length $1$. The correct consensus set ranked $3$ has more test cases (i.e., $222$), while it has a lower consensus score due to the small number of code solutions (i.e., $9$). The second example in Figure 11b shows the most common situation where CODET fails because the model cannot fully understand the problem. We can find that the incorrect solutions in the top $1$ consensus set are totally missing the points of the given problem. While the model still tends to generate more incorrect solutions and test cases based on its wrong understanding. All the bad cases call for future improvements on the quality of generated code solutions and test cases.

```
def prime_length(string):
    """
    Write a function that takes a string and returns True if the string length is a prime number
    or False otherwise
    """
```

Rank #1: The consensus set has **20** solutions and **205** test cases. An example solution:                    **Incorrect**

```
    num_chars = len(string)
    if num_chars == 1:
        return True
    for i in range(2, num_chars):
        if num_chars % i == 0:
            return False
    return True
```

Rank #3: The consensus set has **9** solutions and **222** test cases. An example solution:                    **Correct**

```
    length = len(string)
    if length < 2:
        return False
    for i in range(2, length):
        if length % i == 0:
            return False
    return True
```

(a) Uncovered corner cases.

```
def minSubArraySum(nums):
    """
    Given an array of integers nums, find the minimum sum of any non-empty sub-array of nums.
    """
```

Rank #1: The consensus set has **16** solutions and **102** test cases. An example solution:                    **Incorrect**

```
    if not nums:
        return 0
    total = nums[0]
    min_sum = total
    for i in range(1, len(nums)):
        if total < 0:
            total = nums[i]
        else:
            total += nums[i]
        min_sum = min(min_sum, total)
    return min_sum
```

Rank #2: The consensus set has **7** solutions and **96** test cases. An example solution:                    **Correct**

```
    if not nums:
        return 0
    min_sum = float('inf')
    n = len(nums)
    for i in range(n):
        curr_sum = 0
        for j in range(i, n):
            curr_sum += nums[j]
            min_sum = min(min_sum, curr_sum)
    return min_sum
```

(b) Failure of Problem Understanding.

Figure 11: Two incorrect cases from the HumanEval benchmark, where CODET cannot find the correct consensus sets due to (a) uncovered corner cases, or (b) failure of problem understanding.

