# OpenReview forum: "CodeT:  Code Generation with Generated Tests"
_ICLR.cc/2023/Conference — ICLR 2023 poster_

### Official Review · Reviewer_GQjW · 2022-10-23

**Confidence:** 5
**Correctness:** 3
**Technical Novelty And Significance:** 3
**Empirical Novelty And Significance:** 4
**Recommendation:** 8

**Clarity, Quality, Novelty And Reproducibility:**

The paper is overall very well written and comprehensive. I leave a few minor notes on the writing below.

**Writing Comments:**
- Figure 3: it would be helpful to expand the caption to describe the meaning of both the edges and boxes.
- P4: "this score equals to" -> "this score is equal to"
- P6: "super hard" -> consider using more formal terminology (e.g., "especially hard")
- P9: "in the future work" -> remove "the"


**Strength And Weaknesses:**

This work introduces a relatively straightforward, yet highly effective technique, and explores its design decisions in great depth. I was left with nearly no questions after reading through the results in the appendix, and find the approach convincing. This makes it a strong contribution to the domain of code-generating models.

In terms of weaknesses, a lesser complaint is that this work makes very little effort to connect its findings to the theme of representation learning. Implicit in the experiments and results is the observation that LLMs must have some instrinsic notion of code execution, at least as syntactically represented by test cases, but this is neither explored per se, nor does the work draw connections with any related fields and applications. As such, it seems like it would be a better fit at a software engineering venue. Nevertheless, conferences such as this tend to draw papers from a very wide range of application domains, so this does not change my overall verdict.

Another relatively minor issue is the decision to remove input/output examples from the (HumanEval) prompts, or rather, the discussion around that experimental decision. While (the end of) Sec. 2.1 reasonably states that this helps avoid biasing the models, App. A makes a much stronger claim that removing these cases is required to "fairly" verify CodeT. This strikes me as a rather odd statement: program synthesis research has historically mostly operated under the assumption that input/output examples would be provided by a user, just like the HumanEval benchmark does. The CodeT approach also clearly both benefits from these tests _and_ generalizes beyond them, judging from the substantial performance gains over its own results without "seed" tests as reported in Tab. 4. While it is entirely reasonable to compare both conditions on the premise that not requiring examples involves less effort by the user, suggesting it is in some way more fair seems off. I would suggest reporting both settings side by side in the main text instead. Finally, the text in 2.1 also claims that removing the original tests increases the "diversity and difficulty" of the generated tests, which does not appear to be substantiated by experiments.

**Summary Of The Paper:**

Large language models trained on source code can generate many plausible solutions to programming problems. This work proposes to simultaneously use these models to generate tests for the same prompts, which are then used to re-rank the generated solutions, using a consensus set based algorithm. The results show that this technique substantially improves the correctness of the highest ranked sample(s) across a range of language models, and additionally provide an analysis of the impact of test quality, which underscores that especially powerful models benefit most from this modification because they are also most likely to generate meaningful tests.

**Summary Of The Review:**

This paper is well-written, thorough, and yields strong results. Its proposed technique is not particularly complex, but is both effective and very well investigated. Its main limitation is a relatively narrow focus on a specific programming task approach purely from a large language model perspective, which leaves it both without strong connections to the role of self-evaluation in the broader ML community, and to make the occasional questionable claim relative to prior work in program synthesis.

---

> ### Author Response · Authors · 2022-11-15
> **Response to Reviewer GQjW**
>
> Thank you for your positive feedback and helpful suggestions on our work. We have addressed and improved the writing issues you suggested.
>
> As for removing input/output examples from the prompts, we consider it indispensable.
> (1) It is deceptive to use real test cases as prompts to generate test cases. Models will tend to directly copy the real test cases instead of generating by themselves. Intuitively, this would reduce the "diversity and difficulty" of the test cases. We removed this phrase to make our statement more rigorous.
> (2) While code generation benchmarks (e.g., HumanEval) often include test case examples, test case generation benchmarks do not [1, 2, 3]. In real-world scenarios, creating test cases is time-consuming, which has led to the flourishing of automatic test case generation approaches [1, 2, 3].
>
> **Reference**
>
> [1] Arcuri, Andrea. Many independent objective (MIO) algorithm for test suite generation. https://arxiv.org/pdf/1901.01541.pdf
>
> [2] Panichella, Annibale, Fitsum Meshesha Kifetew, and Paolo Tonella. Automated test case generation as a many-objective optimisation problem with dynamic selection of the targets. https://orbilu.uni.lu/bitstream/10993/30978/1/tse2017.pdf
>
> [3] Fraser, Gordon, and Andrea Arcuri. Evosuite: automatic test suite generation for object-oriented software. https://www.cse.unr.edu/~dascalus/Paper\_MARLON.pdf
>
> Best,
>
> Paper1043 Authors

---

### Official Review · Reviewer_ziee · 2022-10-24

**Confidence:** 4
**Correctness:** 3
**Technical Novelty And Significance:** 3
**Empirical Novelty And Significance:** 3
**Recommendation:** 6

**Clarity, Quality, Novelty And Reproducibility:**

Edit - I have updated my evaluation of quality and originality after reading the review by Reviewer ZjXf.

**Quality and Originality :**

The idea of reusing the same model with a carefully engineered prompt is innovative and yields surprisingly good results for such a (conceptually) simple change. However when presented in the light of the AlphaCode paper, this diminishes the novelty of the approach somewhat. Moreover, this comparison is missing in the paper.

**Clarity :**

I do feel that the presentation of the paper could be vastly improved. High-level comments :

1. There is a lot of text in the paper with numbers embedded in it, which could possibly be better represented as a table/diagram. In particular, I would like one table to which I can refer to see the different datasets, number of problems, average number of "ground truth" tests per problem (see point 1 under "Weaknesses"), and sampling number. That brings me to my next comment...

1. After several re-reads, I still was unable to understand what is meant by *"we sample 100 times for each problem and extract the first five syntactically correct test cases from each sample"*. I was of the impression that we can generate $n$ solutions and $k$ test cases independent of each other. But this sentence seems to suggest that for **each** solution, we generate $5$ test cases. Why would there be a correspondence between generated test cases and generated solutions?

1. (Minor) I have a question about the following step - "If we want to obtain k code solutions, we can select the top k consensus sets with the highest scores, and one code solution is picked up from each of the k consensus sets." Suppose you've generated 100 solutions grouped into 5 consensus sets of 20 solutions each. Then if you pick one from each set, you will have only 5 solutions. How do you obtain 10 solutions to compute pass@10?

**Reproducibility :**

I couldn't find provided code to reproduce the results, but the approach is simple enough that I think one could reproduce the results without unreasonable effort.

**Strength And Weaknesses:**

**Edit :** After reading the review by reviewer ZjXf, I have several new concerns about the paper and evaluation.

**Comparison with AlphaCode:**

The AlphaCode paper uses a separate trained model to generate test inputs, and uses the outputs on these test inputs to cluster its generated solutions (which can be up to 1e4). In comparison, CodeT re-uses the same model to generate test inputs *and* desired outputs. When presented in this light, this diminishes the novelty of CodeT somewhat. A detailed discussion should be included in the paper.

**Computation of pass@k:**

The "biased" approach to compute pass@k is to just generate $k$ solutions and see if any one of them passes all test cases. Using the "unbiased" approach from Chen et al. (2021), one would generate $n$ solutions, and compute the number $c$ that pass all the hidden test cases. Then the pass@k is computed using a combinatorial formula.

As I understand it, the baselines have been evaluated with the "unbiased" formula, but CodeT has been evaluated with the "biased" formula (mentioned under "Implementation Details" in Section 3). This seems a little strange and not kosher. I think this evaluation needs a rethink.

**pass@k vs n@k:**

The metric n@k as presented in the AlphaCode paper is as follows - generate $k$ solutions and use some filtering mechanism to narrow it down to $n$ solutions where $n \le k$. Then if any one of the $n$ solutions passes all the hidden test cases, report success on that problem.

*This seems like the setting that you should be evaluating CodeT on.* In particular, pass@k does not seem well suited to your problem setting, and results in an odd mismatch in evaluation procedures between CodeT and your baselines.

I would reformulate the entire paper as a comparison of *different filtering approaches*, with the evaluation based on the n@k metric. The filtering approaches would be:
1. Primitive filtering (baseline). Just take the top $k$ solutions out of $n$ generated solutions ("biased" pass@k).
1. Random filtering - take a random sample of $k$ solutions out of $n$ generated solutions. This (in expectation) is the "unbiased" pass@k metric defined in Chen et al. (2021).
1. The AlphaCode approach - train another model to generate test inputs, and cluster the $n$ solutions based on their outputs on these generated inputs.
1. CodeT - reuse the same model to generate test inputs as well as desired outputs.

There needs to be a detailed comparison between the latter two approaches. I understand that it is difficult to recreate the AlphaCode setting with their trained input generation model, but at the very least there should be a *qualitative* comparison.

My old review comments (pre-edit) follow.

---------------------------------------------------------------------------------------------------------------

**Strengths :**

1. Simple, elegant method that requires no retraining
1. Convincing performance improvements with thorough experiments

**Weaknesses**

1. This approach could potentially increase the code generation time several-fold, especially if you have ~1e3 tests for 1e2 solutions => ~1e5 executions. The HumanEval benchmark has just 7.7 "ground-truth" tests on average per problem, so if one were to execute all top 100 solutions on the given tests then there would be ~770 executions. I did notice and appreciate the note at the bottom of the Q2 section and the experiments in Appendix G.3, but this should really be its own section. Execution time is, in my opinion, a significant factor in determining the real-world utility of this model. I would ideally like to see :

    * How many "ground-truth" test cases does each benchmark provide, on average, per problem?

    * How many executions (code running on test) does CodeT require, on average, per benchmark?

    * The results in Appendix G.3 moved to the main text.

2. (Minor) The choice of using the *product* of number of solutions and number of test cases seems slightly counter-intuitive to me : why would we (intuitively) care about how many other solutions matched the same test cases? It's all the more intriguing because you say that "such cases are not rare", and your ablation study in Appendix H shows that this is an important design choice. Some intuition here would be helpful.

**Summary Of The Paper:**

This paper tackles the problem of generating a code solution to a programming problem. The key observation is that current models like Codex can generate solutions, but selecting the appropriate solution is challenging. This is reflected in the large gap between pass@100 and pass@1. If test cases were available, one could simply run the top 100 generated solutions on all the test cases to get the best one. But in several real world settings, test cases are not available.

This paper proposes to generate test cases and solutions simultaneously with the same model (no further training necessary). They engineer a prompt that makes the model generate test cases, and use a random sampling strategy to run selected solutions on selected test cases. From the results of these runs, they pick the best solution according to certain criteria.

They show that their approach achieves approximately a 10% improvement on the HumanEval and MBPP benchmarks, with minor improvements on the Apps and CodeContests benchmarks.


**Summary Of The Review:**

**Edit** - I have updated my evaluation and score after reading the review by Reviewer ZjXf.

The idea that this paper presents is simple, elegant and innovative. However it is missing a comparison with the approach taken by AlphaCode, i.e., training a separate model to generate test inputs. Further, the evaluation is shaky and in my opinion, needs to be reformulated around the n@k metric and as a comparison of different filtering approaches. On the whole, I think this is an innovative idea, but the paper in its current form is not ready for publication. I would be happy to reconsider my score if the authors are able to address these concerns.

---

> ### Author Response · Authors · 2022-11-15
> **Response to Reviewer ziee (1/2)**
>
> We appreciate your kind words about the elegance and effectiveness of our approach. We have carefully answered the questions in your comments, focusing on the comparison and metric issues. We hope our answers can resolve your doubts. Please reconsider the review based on our answers.
>
> **1. Comparison with AlphaCode.**
>
> We provide the differences between CodeT Clustering (i.e., dual execution agreement) and AlphaCode Clustering in the responses to Reviewer ZjXf's W2 & W3. Please refer to those responses for details. Here we want to answer your question: *why would we (intuitively) care about how many other solutions matched the same test cases?*.
>
> Our approach is inspired by the RANSAC algorithm, which uses the concept of **consensus set**. A consensus set is a collection of code solutions and test cases that agree on the same functionality. In other words, all the code solutions in a consensus set can pass the same test cases, and all the test cases in a consensus set validate the functionality of the code solutions.
>
> We can measure the quality of a consensus set by its consensus score, which is the product of two factors: (1) the number of code solutions that share the same functionality and (2) the number of test cases that confirm this functionality. As we stated in Section 1, our intuitions are that (1) a code solution that is more consistent with other solutions is more likely to follow the problem specification and (2) a code solution that passes more test cases is more likely to be correct. The first intuition is similar to the idea of self-consistency [1] and AlphaCode Clustering [2]. However, AlphaCode Clustering considers solutions that produce the same test outputs to have the same functionality, regardless of the second factor, i.e., the number of test cases supporting that functionality, because it only uses test inputs.
>
> Alternatively, we can view the consensus score as the size of the consensus set. The size reflects the number of (solution, test) pairs that agree with each other in the consensus set. Our assumption, based on RANSAC, is that correct behaviors tend to agree with each other more often than incorrect ones, which are more diverse. Therefore, using the size of the consensus set as a score makes sense, as it indicates the amount of agreement among the behaviors.
>
> **2. About the metric of pass@k.**
>
> Please refer to the answer to ZjXf' W1 about the explanation of pass@$k$ (with $n$ samples).  We have made this more clear in the revision.
>
> **3. Numbers are better represented as a table/diagram.**
>
> Thanks for your suggestion. We added a separate table to clearly show the statistics of different benchmarks (Table 1 in the revised paper), as shown below. The execution time is important, which motivated us to conduct the experiments with fewer test cases (Appendix G.3 in the original paper). We took your suggestion and moved it to the main text (Section 4.3 Q3 in the revised paper).
>
> | Benchmark | Problems | GT Tests | $n$ |
> |-----------|----------|----------|----------|
> | HumanEval   | 164     | 7.77   | 100   |
> | MBPP          | 427     | 3.1    | 100   |
> | APPS          | 5,000   | 20.99  | 50    |
> | CodeContests  | 165   | 203.7  | 1,000 |
>
> Table: Statistic numbers of different benchmarks: the total number of problems in the benchmark (*Problems*), the average number of ground-truth test cases per problem (*GT Tests*) and the number of sampling code solutions for each problem ($n$). Specially, APPS includes $1,000$ introductory, $3,000$ interview, and $1,000$ competition level coding problems.

---

> > ### Comment · Reviewer_ziee · 2022-12-05
> > **Response to authors**
> >
> > I appreciate that the authors have taken steps to increase the clarity of the paper and address some of the concerns raised by the reviewers. In particular, I appreciate that they attempted to reproduce the clustering method followed by AlphaCode. Here are my thoughts regarding the central issue at hand, i.e., the metric pass@k.
> >
> > * In this field, there are many terms like pass@k, n@k, ranked pass@k, pass@k with n, etc., sometimes with multiple definitions for a single term (like pass@k). This is confusing at best and deliberately obscure at worst. There needs to be an attempt in the field at unifying/clarifying these metrics. This paper is unfortunately caught up in this ecosystem of confusion.
> >
> > * In practical terms (without relying on definitions and terms), this is the problem I see. This paper presents a strategy for choosing the best $k$ solutions out of $n$ **ranked** solutions. This is compared with a baseline of **randomly** choosing $k$ solutions out of the same $n$ solutions. But it is not compared with a baseline of simply choosing the top $k$ solutions in the **existing ranking**.
> >
> > * It can be thought of like this - if CodeT simply preserved the same top $k$ without doing any re-ranking, would it be better or worse than the baseline? With the current evaluation, we don't know!
> >
> > * The Codex paper (Chen et al. (2021)) claims that choosing the top $k$ solutions has too high variance, so they define a low-variance estimator, that is, randomly choosing $k$ solutions out of $n$. They call this "unbiased", but according to my understanding, that is only true if the $n$ solutions are not ranked. Maybe for their specific model, the exact ranking within the top $n$ is unreliable ("high variance"), so it makes the estimator effectively unbiased. I don't like this definition, however it is a precedent that has been set.
> >
> > The metrics section is now a little clearer after the revision, and in particular the sentence "We use the unbiased definition of pass@k as our baseline (Chen et al., 2021), where k solutions are randomly picked from n samples" is very helpful. For now, I will raise my score to 5 pending further internal discussion with the other reviewers.

---

> > > ### Author Response · Authors · 2022-12-07
> > > **Response to Reviewer ziee**
> > >
> > > Thank you for your feedback. We are grateful to see that some of your concerns have been addressed.
> > >
> > > Regarding the pass@$k$ metric, we would like to re-emphasize that our goal is clear, which is to ultimately recommend $k$ code solutions to users, regardless of the method used. Hence, randomly picking and using ranking can be compared fairly, which is consistent with other related works[1,2,3]. Furthermore, we do not have access to the **existing ranking** models or data[1,2], so a fair comparison is challenging. We have tried our best to add the ranking method used by AlphaCode as a baseline.
> > >
> > > > if CodeT simply preserved the same top $k$ without doing any re-ranking, would it be better or worse than the baseline?
> > >
> > > The pre-trained language model randomly and independently samples $n$ code solutions, which are not ranked. Hence, selecting top $k$ without doing any re-ranking is the same as our baseline (selecting randomly).
> > >
> > > To our understanding, the raw pass@$k$ is to generate only $k$ code solutions, which would have high variance[4]. Hence, the Codex paper[4] suggests using the approach of randomly choosing $k$ solutions out of $n$, which is reasonable and can be used as our baseline.
> > >
> > > We hope our answers have resolved your concerns. If so, we would appreciate it if you could increase the rating. Thanks again for your constructive feedback and suggestions.
> > >
> > >
> > >
> > > **Reference**
> > >
> > > [1] Inala, Jeevana Priya, et al. Fault-Aware Neural Code Rankers. https://arxiv.org/pdf/2206.03865.pdf
> > >
> > > [2] Key, Darren, Wen-Ding Li, and Kevin Ellis. I Speak, You Verify: Toward Trustworthy Neural Program Synthesis. https://arxiv.org/pdf/2210.00848.pdf
> > >
> > > [3] Lahiri, Shuvendu K., et al. Interactive Code Generation via Test-Driven User-Intent Formalization. https://arxiv.org/pdf/2208.05950.pdf
> > >
> > > [4] Chen, Mark, et al. Evaluating large language models trained on code. https://arxiv.org/pdf/2107.03374.pdf

---

> > > > ### Comment · Reviewer_ziee · 2022-12-07
> > > > **Existing ranking**
> > > >
> > > > > The pre-trained language model randomly and independently samples $n$ code solutions, which are not ranked.
> > > >
> > > > What about the likelihood assigned by the language model to each sample? This is usually interpreted as a "confidence score" of a sequence prediction, like in beam search.
> > > >
> > > > For example in the Codex paper [1], they rank the predictions by the mean log probability of each token. They show that that is better than random ranking by up to 11.6% (Section 4.5).
> > > >
> > > > [1] Chen, Mark, et al. "Evaluating large language models trained on code." arXiv preprint arXiv:2107.03374 (2021).

---

> > > > > ### Author Response · Authors · 2022-12-08
> > > > > **Response to Reivewer ziee: Existing Ranking**
> > > > >
> > > > > Yes, the Codex paper [1] uses the likelihood values for ranking. From Figure 7 \& 10 of the Codex paper [1], the conclusion is that, **when the temperature is fixed to 0.8**, it is better to choose a solution with the highest mean likelihood than choose it randomly. However, according to Figure 5 of the Codex paper [1] and Figure 9 of our revised paper, temperature matters a lot when generating code. In general, lower temperature leads to lower diversity and higher pass@1, closer to greedy search. To make a fair comparison, we used a very low temperature to get a very competitive baseline pass@1, as described in our implementation details (Appendix A in the revised version).
> > > > >
> > > > > Recently (just a few days ago), Zhang et al. [2] conducted a thorough experiment using likelihood to solve the same task as we do. According to their results, selecting the code solution with the highest sum likelihood has similar pass@1 scores to a random selection. Normalized likelihood works a litter better, but still inferior to CodeT.
> > > > >
> > > > >
> > > > >
> > > > > **Reference**
> > > > >
> > > > > [1] Chen, Mark, et al. Evaluating large language models trained on code. https://arxiv.org/pdf/2107.03374.pdf
> > > > >
> > > > > [2] Zhang, Tianyi, et al. Coder Reviewer Reranking for Code Generation. https://arxiv.org/pdf/2211.16490.pdf

---

> > > > > > ### Comment · Reviewer_ziee · 2022-12-08
> > > > > > **Temperature is indeed important**
> > > > > >
> > > > > > Thanks for that clarification, that makes sense. Sampling temperature is an important factor that I initially didn't take into account. Although the experiment with log likelihood based ranking should ideally be part of this paper, in light of the sampling temperature difference it is no longer a major concern of mine. I have accordingly increased my score to 6.

---

> > > > > > > ### Author Response · Authors · 2022-12-09
> > > > > > > **Response to Reviewer ziee: Thank You**
> > > > > > >
> > > > > > > We are very grateful that you consider our work more positive. We would like to thank you again for your valuable comments and constructive suggestions.

---

> ### Author Response · Authors · 2022-11-15
> **Response to Reviewer ziee (2/2)**
>
> **4. Why would there be a correspondence between generated test cases and generated solutions**
>
> We do not generate test cases for each solution, but rather for each problem. To clarify, when generating test cases for each problem, we generate $100$ samples of code snippets, each of which may contain multiple assertion statements. These assertion statements are potential test cases, but we do not use all of them. Instead, we extract only the first five syntactically correct assertion statements from each sample, and discard the rest. This way, we obtain at most $500$ test cases per problem, depending on the quality and quantity of the assertion statements in each sample.
>
> Hence, in the sentence "we sample 100 times for each problem and extract the first five syntactically correct test cases from each **sample**", a **sample** here means a code snippet generated by the model, which may contain multiple assertion statements. We have made this clearer in our revision.
>
> **5. How do you obtain $k$ solutions from $<k$ consensus sets?**
>
> CodeT has a rolling solution selection approach. Specifically, the selection is done from the highest scoring consensus set to the lowest. When reaching the set with the lowest score, it repeats from the highest scoring consensus set. In most cases, the number of consensus sets is larger than $k$, as shown in Figure 6. We added this in Appendix A in the revision.
>
> BTW: The comment mentions "I couldn't find provided code to reproduce the results". In fact, we have submitted all the source code and data to the supplementary material, which will be made public later.
>
> **Reference**
>
> [1] Wang, Xuezhi, et al. Self-consistency improves chain of thought reasoning in language models. https://arxiv.org/pdf/2203.11171.pdf.
>
> [2] Li, Yujia, et al. Competition-level code generation with alphacode. https://arxiv.org/pdf/2203.07814.pdf
>
>
> Best,
>
> Paper1043 Authors

---

> ### Author Response · Authors · 2022-11-23
> **Gentle Reminder to Reviewer ziee**
>
> Thank you again for your valuable comments.
>
>
> From our perspective, we feel that we have clarified all of the concerns that you raised in the initial review. We would appreciate it if you could re-evaluate our paper in light of our response and find our work more positive! If you have any further questions, we are always available for discussion.
>
>
> Best,
>
> Paper1043 Authors

---

### Official Review · Reviewer_ZjXf · 2022-10-24

**Confidence:** 5
**Correctness:** 2
**Technical Novelty And Significance:** 2
**Empirical Novelty And Significance:** 1
**Recommendation:** 5

**Clarity, Quality, Novelty And Reproducibility:**

Clarity:

The paper evaluates based on $pass@k$ when it really should be evaluating on $n@k$ for a fair comparison. This aspect is not made clear at all.

Originality:

Generating test cases using language models has been explored in several papers. Clustering generated solutions based on generated inputs was explored in the AlphaCode paper. Clustering generated solutions based on binary success/failure on generated *unit tests* (which include the expected output) has not been explored but there is no motivation given as to why this might work better than the previous AlphaCode work -- nor is any comparison done.

**Strength And Weaknesses:**

While I think the paper is the start of an interesting direction, I think in its current form, it is lacking important comparisons as well as motivation.

In the paper, they claim to compare on $pass@k$ (Table 1) where they set $k=1,2,10$. However, $pass@k$ is relevant when *all* the models are allowed to generate only upto $k$ solutions. In this instance however, CodeT is allowed to generate upto 100 solutions (from which 1,2,10 are selected) but other models are only allowed to generate 1,2,10 solutions. Thus the CodeT model is allowed a far larger computation budget and has an unfair advantage under the $pass@k$ metric.

For such a scenario, it is the $n@k$ metric from the AlphaCode paper that makes the most sense and provides for a fair comparison -- where they also generate $k$ solutions but then select $n$ solutions out of those to evaluate against the ground truth unit tests. In the AlphaCode paper, they select the $n$ representative solutions by executing the $k$ solutions on a model trained to generate inputs and clustering based on the output.

In this paper, they also do a clustering -- however instead of clustering by executing on a model trained to generate inputs and clustering based on the output, the clustering is based on executing on the code generation language model based generated inputs and the binary evaluation of whether the test case passes or not.

Given the above, the main two issues that need to be worked on are as follows

(1) How does this paper's approach compare with AlphaCode's approach for clustering and then evaluating the selected solutions?
(2) Why might clustering based on the binary feedback of success or failure from generated test cases from a language model perform better than a more fine grained clustering based on the generated output on an input?

**Summary Of The Paper:**

The paper is part of the recent stream of papers which use large language models to generate code to solve coding contest type problems. Two popular ways of measuring the performance of such models is to (1) evaluate $pass@k$ -- $k$ solutions are generated for a problem and evaluated against the unit tests for that problem. If even 1 solution clears all the test cases for that problem, then the problem is considered solved. The metric is the fraction of problems so solved in the dataset (2) $n@k$ -- $k$ solutions are generated but *only* $n$ solutions are evaluated against the unit tests for the problem. If even 1 out of the $n$ (instead of $k$) solutions passes all the test cases, the problem is considered solved and the metric is the fraction of problems so solved. This metric was widely used in the AlphaCode paper [1]

The idea behind this paper is as follows. Given a prompt, they first generate $K$ solutions to the problem that the prompt describes. In their experiments $K=100$. They then use the language model to generate new test cases $T$ for the problem. While the validity of any of the test cases in $T$ is unknown, the agreement of the different generated solutions in terms of passing or not passing test cases in $T$ can be used to *cluster* the solutions.

They then use this clustering to select $n=1,2,10$ solutions from the $K=100$ candidate solutions they generate. They compare this against baseline models which are allowed to generate only $n=1,2,10$ solutions. They claim improvements for $n=1,2,10$ but not for $n=100$ when the computation budget is the same for the baseline as well as their model.

[1] Competition-Level Code Generation with AlphaCode (https://arxiv.org/abs/2203.07814)

**Summary Of The Review:**

The reasons for my rating are

(1) $pass@k$ is an unfair metric for what they're trying to evaluate -- as they're selecting a subset of generated solutions which gives their model effectively a far larger computation budget than their baselines.
(2) No comparison against AlphaCode style clustering techniques for solution subset selection.
(3) Motivation for why their clustering may work better than AlphaCode's clustering is not given.

**Edit**: After reading the author's response, I'm raising my score to a 5. My response (copy-pasted from below)

"I thank the authors for their detailed response as well as the additional experiments.

I still disagree that raw pass@k is an appropriate metric as you're comparing a method with filtering to a method without. That would suffice if there were no extant filtering or ranking methods but that is not the case.

I agree that ranked pass@k is appropriate. I also find the comparison with AlphaCode critical and that it shows an advantage for CodeT is encouraging and a point in CodeT's favor.

Overall however, I agree with Reviewer ziee that this should have been presented as a filtering paper -- which would change the narrative a great deal as well as would require more thorough experiments -- for example there are several reranking methods that could be used to filter out top-k solutions and could serve as good baselines.

I encourage the authors to modify the paper and add such baselines.

I'll raise my score to a 5 but I'm not comfortable voting for an accept given that the paper should be framed as a filtering paper and should involve a much more thorough investigation against various alternative baselines for filtering/ranking "

---

> ### Author Response · Authors · 2022-11-15
> **Response to Reviewer ZjXf (1/2)**
>
> We appreciate your feedback, but we have some different perspectives on the comments. We provide more explanations for each weakness you pointed out in your comments. We hope you can reconsider the review after reading our responses.
>
> **W1. pass@$k$ is an unfair metric for what they're trying to evaluate.**
>
> We do not agree on this. The main purpose of pass@$k$ is to measure the likelihood that at least one of the $k$ solutions generated by the system can pass the test cases for a programming problem in the test set. This reflects the usefulness of the system for users who can choose from multiple solutions suggested by the system. There are various ways to estimate this likelihood under different settings.
>
> When we use pass@$k$, we mean pass@$k$ with $n$ samples. That is, for each problem, we sample $n$ solutions and then present $k$ of them to users. The Codex paper [1] uses an unbiased definition of pass@$k$ that randomly picks $k$ samples from $n$ solutions for output. However, our CodeT uses a dual execution agreement mechanism to select $k$ samples from $n$ solutions, which does not change the meaning of pass@$k$ with $n$ -- selecting $k$ from $n$ samples for output. Therefore, we claim that these results are still comparable.
>
> Note: In papers on similar tasks, authors may use different notations, such as $n$@$k$ [2] and ranked pass@$k$ [3]. This may cause some confusion. To address this, we revised the metric section of our paper to make it more clear to the readers.
>
> **W2. No comparison against AlphaCode style clustering techniques for solution subset selection.**
>
> We appreciate the suggestion, but AlphaCode Clustering relies on a separate test input generation model, which needs extra training and annotation. The model is unavailable and hard to replicate, as the paper does not provide sufficient details. We implemented an adapted version of AlphaCode Clustering using the test inputs from the test cases generated by CodeT. We ran all candidate code solutions on the test inputs, and grouped them by outputs. The clusters were ranked by size. The table below shows the results. As seen, AlphaCode Clustering is inferior to CodeT, again demonstrating the effectiveness of CodeT. We have updated the paper with new results and discussion.
>
>
> | Models |  | Baseline |   | |   | AlphaCode |  | |  | CodeT | |
> |------------|:----------:|:-----------:|:-----------:|:-----------:|:-----------:| :-----------:|:----------:|:-----------:| :----------:|:----------:|:-----------:|
> | pass@$k$ | $1$ | $2$ | $10$ | | $1$ | $2$ | $10$ | | $1$ | $2$ | $10$ |
> | *code-cushman-001* |  33.5 |  54.3 |  77.4 | | 39.6 |  46.4 |  63.8 | | **44.5** |  **50.1** |  **65.7** |
> | *code-davinci-001* |  39.0 | 60.6 |  84.1 | |  41.6 |  50.7 |  75.6 | |  **50.2** |  **58.9** |  **75.8** |
> | *code-davinci-002* |  47.0 | 74.9 |  92.1 | |  55.1 |  64.1 |  84.4 | |  **65.8** |  **75.1** |  **86.6** |
>
> Table: Pass@$k$ (%) on the HumanEval benchmark.

---

> > ### Comment · Reviewer_ZjXf · 2022-11-18
> > **Response to authors**
> >
> > I thank the authors for their detailed response as well as the additional experiments.
> >
> > I still disagree that raw pass@k is an appropriate metric as you're comparing a method with filtering to a method without. That would suffice if there were no extant filtering or ranking methods but that is not the case.
> >
> > I agree that ranked pass@k is appropriate. I also find the comparison with AlphaCode critical and that it shows an advantage for CodeT is encouraging and a point in CodeT's favor.
> >
> > Overall however, I agree with Reviewer ziee that this should have been presented as a filtering paper -- which would change the narrative a great deal as well as would require more thorough experiments -- for example there are several reranking methods that could be used to filter out top-k solutions and could serve as good baselines.
> >
> > I encourage the authors to modify the paper and add such baselines.
> >
> > I'll raise my score to a 5 but I'm not comfortable voting for an accept given that the paper should be framed as a filtering paper and should involve a much more thorough investigation against various alternative baselines for filtering/ranking

---

> > > ### Author Response · Authors · 2022-11-19
> > > **Response to Reviewer ZjXf**
> > >
> > > We are grateful for your quick feedback and your review of our paper.
> > >
> > > We do not concur with the idea of presenting our paper as a filtering paper only, because our main contribution is to exploit the potential of large pre-trained language models for self-validation without any additional training or data. We cannot compare our approach with those that rely on extra resources (model training, additional data, or human efforts) [1,2,3,4] in our setting. Moreover, we do not have access to the existing ranking models or data [1,2], so a fair comparison is challenging.
> > >
> > > Regarding the pass@$k$ metric, we have clarified in our revised paper that we use "pass@$k$ with $n$" instead of the raw pass@$k$ metric, which is equivalent to $n$@$k$ and ranked pass@$k$ in theory. We also justify that the comparison between CodeT and the top-1 or random filtering baseline is reasonable, which is consistent with other related works [1, 2, 3].
> > >
> > > We have provided more explanations and references in our paper to support our arguments. We hope our answers have resolved some of your concerns. If so, we would appreciate it if you could revise your review or increase the rating. Thanks again for your constructive feedback and suggestions.
> > >
> > >
> > > **Reference**
> > >
> > > [1] Inala, Jeevana Priya, et al. Fault-Aware Neural Code Rankers. https://arxiv.org/pdf/2206.03865.pdf
> > >
> > > [2] Key, Darren, Wen-Ding Li, and Kevin Ellis. I Speak, You Verify: Toward Trustworthy Neural Program Synthesis. https://arxiv.org/pdf/2210.00848.pdf
> > >
> > > [3] Lahiri, Shuvendu K., et al. Interactive Code Generation via Test-Driven User-Intent Formalization. https://arxiv.org/pdf/2208.05950.pdf
> > >
> > > [4]  Shi, Freda, et al. Natural Language to Code Translation with Execution. https://arxiv.org/pdf/2204.11454.pdf

---

> ### Author Response · Authors · 2022-11-15
> **Response to Reviewer ZjXf (2/2)**
>
> **W3. Motivation for why their clustering may work better than AlphaCode's clustering is not given.**
>
> We argue that CodeT is more original and effective than AlphaCode's clustering, based on the following reasons.
>
> (1) AlphaCode Clustering only uses test inputs, while CodeT uses test cases (input \& output). Test inputs are not enough to distinguish code correctness, while test cases can provide more feedback, as a test case can be correct or incorrect for a given programming problem. CodeT Clustering and AlphaCode Clustering share a similar goal: to cluster code solutions based on code functionality. However, CodeT is more sophisticated because it exploits more information from test cases: it considers both the number of solutions with the same functionality and the number of test cases supporting the functionality. As we explained in Methodology (Section 2.2), each consensus set contains multiple pairs of code solutions and test cases. The consensus score is based on the size of the set, which reflects the number of test cases. The rationale is that the more test cases that agree on a functionality, the more likely this functionality is correct. Our experimental results (Table 2 in the revised paper) and ablation study (Table 13 in the revised paper) demonstrate that accounting for the impact of test cases is crucial. But AlphaCode Clustering cannot achieve this with only test inputs. Therefore, AlphaCode Clustering is similar to the ablated version of CodeT Clustering that ignores the impact of test cases, i.e., setting of *Code Solution Only* $f^{\prime}$ in our ablation study (Appdenix I in revised paper).
>
> (2) Unlike previous work that relied on heuristic rules (e.g., EvoSuite [4] and MOSA [5]) or model training (e.g., AlphaCode [2] and ATHENATES [6]) to generate tests, CodeT leverages the same language model as the code generation to produce test cases in a zero-shot manner. This is a significant innovation, as our approach is very simple and does not require additional data and training. In addition, since we use the same language model to generate code solutions and test cases, the model behaves consistently and conforms to our assumption: correct behaviors tend to agree with each other with a high probability, while incorrect ones tend to be diverse. This ensures that our dual execution agreement can be used as a statistical heuristic to find the best solution. CodeT also enables a more general scenario: natural language problem descriptions, code solutions, test cases, and other kinds of specification, all of which can be used to express the intent of a single task. Thus, we can have them validate each other.
>
> **Reference**
>
> [1] Chen, Mark, et al. Evaluating large language models trained on code. https://arxiv.org/pdf/2107.03374.pdf
>
> [2] Li, Yujia, et al. Competition-level code generation with alphacode. https://arxiv.org/pdf/2203.07814.pdf
>
> [3] Inala, Jeevana Priya, et al. Fault-Aware Neural Code Rankers. https://arxiv.org/pdf/2206.03865.pdf
>
> [4] Fraser, Gordon, and Andrea Arcuri. Evosuite: automatic test suite generation for object-oriented software. https://doi.org/10.1145/2025113.2025179
>
> [5] Panichella, Annibale, Fitsum Meshesha Kifetew, and Paolo Tonella. Reformulating branch coverage as a many-objective optimization problem. https://doi.org/10.1109/ICST.2015.7102604
>
> [6] Tufano, Michele, et al. Unit Test Case Generation with Transformers and Focal Context. https://arxiv.org/pdf/2009.05617.pdf
>
>
> Best,
>
> Paper1043 Authors

---

### Official Review · Reviewer_tYm4 · 2022-10-25

**Confidence:** 4
**Correctness:** 4
**Technical Novelty And Significance:** 3
**Empirical Novelty And Significance:** 3
**Recommendation:** 8

**Clarity, Quality, Novelty And Reproducibility:**

This paper is well-writtern and is of high quality. The proposed method is novel. And I expect that the results should be easy to reproduce.

**Strength And Weaknesses:**

Strength:
- The paper is well-written. The method is well explained with examples. Experiment results are extensive with analysis.
- The proposed method does not rely on any additional input (test cases in particular) besides the problem description/prompts, so it is a self-contained method without much assumption.

Weakness:
- The apparent weakness is obvious and also mentioned in the paper. CodeT method only works for programming problem datasets with clear test cases. It limits the applicability of CodeT, for example, it cannot be used in general code generation scenarios in code companion tools like Copilot.
- Considerring the goal for the proposed method is to prompt or post-processing of 100 generated sample, it makes sense to compare the proposed method with language model generation post-reranking  method, which also intend to select/rank from a number of generations.

**Summary Of The Paper:**

This paper proposes a method to improve pretrained language models' performance on executiion-based benchmarks by using the same model to generate test cases and perform dual execution agreement. The proposed method is simple and effective without any additional training. Experiment results with multiple pretrained models on multiple execution benchmarks show impressive improvement.

**Summary Of The Review:**

Overall, I think this is a good paper with novel method proposed and well-performed experiments.

---

> ### Author Response · Authors · 2022-11-15
> **Response to Reviewer tYm4**
>
> We are grateful for your positive feedback and recognition of our work. We address the weaknesses you pointed out as follows. We hope this can resolve your doubts.
>
> **W1. CodeT method only works for programming problem datasets with clear test cases.**
> It is true that CodeT depends on test cases, but it can handle situations that are very common in real-world applications, such as solving programming problems [1, 2] and ensuring software reliability [3, 4]. Moreover, CodeT opens up a broader possibility: natural language problem descriptions, code solutions, test cases, and other kinds of specification, all of which can express the intention of a single task. We can use them to verify each other. This is an interesting direction for our future work.
>
> **W2. It makes sense to compare with other post-reranking method.**
> We have already tried some post-reranking methods. In the ablation study (Appendix H in the original paper), *Code Solution Only $f^{\prime}$* uses the idea of self-consistency [5] to choose the code solutions with the most consistent functionality. And *Test Case Only* $f^{\prime\prime}$ ranks the solutions by the number of test cases they pass. In addition, we reproduced a simplified version of AlphaCode Clustering [2] to further show the advantage of CodeT. The results are included in our revised paper.
>
> **Reference**
>
> [1] Chen, Mark, et al. Evaluating large language models trained on code. https://arxiv.org/pdf/2107.03374.pdf
>
> [2] Li, Yujia, et al. Competition-level code generation with alphacode. https://arxiv.org/pdf/2203.07814.pdf
>
> [3] Arcuri, Andrea. Many independent objective (MIO) algorithm for test suite generation. https://arxiv.org/pdf/1901.01541.pdf
>
> [4] Panichella, Annibale, Fitsum Meshesha Kifetew, and Paolo Tonella. Automated test case generation as a many-objective optimisation problem with dynamic selection of the targets. https://orbilu.uni.lu/bitstream/10993/30978/1/tse2017.pdf
>
> [5] Wang, Xuezhi, et al. Self-consistency improves chain of thought reasoning in language models. https://arxiv.org/pdf/2203.11171.pdf.
>
> Best,
>
> Paper1043 Authors

---

### Author Response · Authors · 2022-11-15
**General Response: Summary of the Revision**

We thank all reviewers for their thorough and helpful feedback. We have addressed each of the comments in detail and hope to clarify any doubts or concerns about our work. We are happy to discuss further if there are any questions.

Moreover, we have revised the paper and upload a new version. The revised paper includes the following changes:
- Report the results of AlphaCode Clustering as suggested by Reviewer `tYm4`, Reviewer `ZjXf` and Reviewer `ziee`.
- Provide more explanation of the pass@$k$ metric as requested by Reviewer `ZjXf` and Reviewer `ziee`.
- Add a table to show statistical numbers of different benchmarks as recommended by Reviewer `ziee`.
- Move Appendix G.3 in the original paper to the main text as commented by Reviewer `ziee`.
- Fix the writing issues pointed out by Reivewer `GQjW`.


Best,

Paper1043 Authors

---

### Decision · Program_Chairs · 2023-01-20

**Decision:**

Accept: poster

**Justification For Why Not Higher Score:**

Existing issues including the lack of a likelihood-based baseline and the positioning of this work against prior work.

**Justification For Why Not Lower Score:**

The proposed idea of the paper is simple, and the performance of the approach is good on a number of benchmarks, showing its value and passing the threshold for acceptance.

**Metareview: Summary, Strengths And Weaknesses:**

This paper presents an approach to improve the code generation performance of large language models, by generating tests and then clustering the generated code with these tests and then selecting clusters based on both the size of the cluster as well as how well aligned the cluster’s outputs are with the generated test output.  The proposed idea is simple and easily applicable to many code generation tasks, with good results.

The reviewers overall favored acceptance after discussion.  The simplicity of the idea and the good performance improvement on a variety of benchmarks justifies acceptance.  The added comparison against AlphaCode in particular helped put the contribution of this paper in context.

There were a few things raised in the discussion, however, that the authors could consider to improve the paper further, in particular:
1. An obvious baseline to use for reranking / selecting samples is to use the likelihood of the sampled program under the language model itself.  This baseline comes almost for free and can better demonstrate the advantage of the clustering approach.
2. The idea of clustering and selecting clusters based on cluster size is not new.  Compared to the clustering approach described in AlphaCode, this paper only differs in (1) using a single model to do generation of programs as well as generation of test cases, and (2) use both cluster size and consistency of the cluster’s program outputs with generated test outputs for scoring.  From the text of this paper the differences are entirely not clear, and the authors should consider doing a better job at this.
3. The proposed approach was compared with AlphaCode clustering on HumanEval and MBPP, but not on the more challenging APPS and CodeContests datasets, raising a few questions.

**Note From Pc:**

if the above contains the word "oral" or "spotlight" please see: "oral" presentation means -> notable-top-5% and "spotlight" means -> notable-top-25%. As stated in our emails, we are disassociating presentation type from AC recommendations

**Summary Of Ac-Reviewer Meeting:**

The AC and 3 reviewers did an online discussion, the one other reviewer could not attend but send in comments in writing ahead of the discussion.

The discussion focused on 3 common concerns:
1. The pass@k metric: even though this was raised in the reviews, it was decided to be not a significant concern.  But the authors should consider making this clear from the beginning.
2. The position against AlphaCode’s clustering approach.  The reviewers acknowledged the similarities between the approaches, but are not particularly concerned.  The reviewers appreciated the authors’ efforts to add a comparison against AlphaCode’s clustering approach.  The simplicity of the additional ingredients introduced in this paper and the improved performance over AlphaCode’s clustering justified this paper’s value.
3. The lack of a likelihood-based sample selection baseline.  This baseline comes for free from the language model used to generate samples, and is easy to implement.  The reviewers suggested that this should be added as a baseline but believe the proposed clustering approach would still outperform this likelihood-based baseline.